# Acid-base transporters and pH dynamics in human breast carcinomas predict proliferative activity, metastasis, and survival

Nicolai J Toft[1], Trine V Axelsen[1], Helene L Pedersen[2], Marco Mele[3], Mark Burton[4,5], Eva Balling[3], Tonje Johansen[2], Mads Thomassen[4,5], Peer M Christiansen[3,6], Ebbe Boedtkjer[1]*

[1]Department of Biomedicine, Aarhus University, Aarhus, Denmark; [2]Department of Pathology, Regionshospitalet Randers, Randers, Denmark; [3]Department of Surgery, Regionshospitalet Randers, Randers, Denmark; [4]Department of Clinical Genetics, University of Southern Denmark, Odense, Denmark; [5]Clinical Genome Center, University and Region of Southern Denmark, Odense, Denmark; [6]Department of Plastic and Breast Surgery, Department of Clinical Medicine, Aarhus University Hospital, Aarhus, Denmark

**Abstract** Breast cancer heterogeneity in histology and molecular subtype influences metabolic and proliferative activity and hence the acid load on cancer cells. We hypothesized that acid-base transporters and intracellular pH ($pH_i$) dynamics contribute inter-individual variability in breast cancer aggressiveness and prognosis. We show that $Na^+,HCO_3^-$ cotransport and $Na^+/H^+$ exchange dominate cellular net acid extrusion in human breast carcinomas. $Na^+/H^+$ exchange elevates $pH_i$ preferentially in estrogen receptor-negative breast carcinomas, whereas $Na^+,HCO_3^-$ cotransport raises $pH_i$ more in invasive lobular than ductal breast carcinomas and in higher malignancy grade breast cancer. HER2-positive breast carcinomas have elevated protein expression of $Na^+/H^+$ exchanger NHE1/SLC9A1 and $Na^+,HCO_3^-$ cotransporter NBCn1/SLC4A7. Increased dependency on $Na^+,HCO_3^-$ cotransport associates with severe breast cancer: enlarged $CO_2/HCO_3^-$-dependent rises in $pH_i$ predict accelerated cell proliferation, whereas enhanced $CO_2/HCO_3^-$-dependent net acid extrusion, elevated NBCn1 protein expression, and reduced NHE1 protein expression predict lymph node metastasis. Accordingly, we observe reduced survival for patients suffering from luminal A or basal-like/triple-negative breast cancer with high *SLC4A7* and/or low *SLC9A1* mRNA expression. We conclude that the molecular mechanisms of acid-base regulation depend on clinicopathological characteristics of breast cancer patients. NBCn1 expression and dependency on $Na^+,HCO_3^-$ cotransport for $pH_i$ regulation, measured in biopsies of human primary breast carcinomas, independently predict proliferative activity, lymph node metastasis, and patient survival.

*For correspondence: eb@biomed.au.dk

## Introduction

Breast cancer heterogeneity is a challenge in clinical practice and calls for extensive patient stratification. Similarly, the underlying tumor biology needs evaluation in stratified patient populations. Breast cancers classify into five molecular subtypes (normal-like, luminal A and B, HER2-enriched, basal-like) that differ in metabolic and proliferative activity, metastatic potential, therapeutic responsiveness, and prognosis (*Dai et al., 2015*). Whereas radical surgery can cure most patients with localized breast cancer, disseminated disease requires additional systemic therapy. Available successful

therapies target the receptor tyrosine-protein kinase HER2 (ErbB2/neu) and estrogen receptors. However, patients with basal-like/triple-negative breast cancer lack targeted treatment options and currently receive classical chemotherapy (e.g., anthracycline and taxane in combination) with considerable adverse effects (*Ávalos-Moreno et al., 2020*).

Accelerated intermediary metabolism in breast cancer tissue (*Voss et al., 2020*) burdens the molecular pathways for acidic waste product elimination. In solid cancer tissue, extracellular pH ($pH_o$) can reach as low as 6.5 (*Voss et al., 2020*; *Vaupel et al., 1989*), which is distinct from corresponding normal tissue with $pH_o$ around 7.3–7.4. The acidity of the extracellular tumor microenvironment challenges intracellular pH ($pH_i$) homeostasis as it inhibits cellular net acid extrusion (*Bonde and Boedtkjer, 2017*). The changes in metabolic profile and proliferative rate of cancer cells contribute to the acidity of the tumor microenvironment, are important determinants of the malignant phenotype, and shape breast cancer progression (*Parks et al., 2017*; *Persi et al., 2021*; *Boedtkjer and Pedersen, 2020*).

Cells generally eliminate their metabolic acid load via membrane proteins that mediate $H^+$ extrusion (e.g., $Na^+/H^+$ exchange, $H^+$-ATPase activity) or $HCO_3^-$ uptake (e.g., $Na^+,HCO_3^-$ cotransport) (*Aalkjaer et al., 2014*; *Xu et al., 2018*; *Stransky et al., 2016*; see *Figure 1B*). Tissue relying partly on fermentative glycolysis can also eliminate acidic waste products from metabolism through coupled transport of $H^+$ and lactate via monocarboxylate transporters (*Pérez-Escuredo et al., 2016*). In human and murine breast cancer tissue analyzed without stratification by molecular subtype, $Na^+$, $HCO_3^-$ cotransport activity is elevated and protein expression of the $Na^+,HCO_3^-$ cotransporter NBCn1 (SLC4A7) and monocarboxylate transporters MCT1 (SLC16A1) and MCT4 (SLC16A3) are upregulated compared to normal breast tissue (*Boedtkjer, 2019*; *Boedtkjer et al., 2013*; *Lee et al., 2016*; *Lee et al., 2018*; *Lee et al., 2015*). Protein expression of the $Na^+/H^+$ exchanger NHE1 (SLC9A1) is more variable in primary breast cancer tissue showing unchanged or only moderately elevated levels (*Lee et al., 2016*; *Lee et al., 2015*) when compared to normal breast tissue as one unstratified group.

Net acid extrusion from cancer cells elevates the cytosolic pH and acidifies the outer cell surface and interstitial space. In various model systems, acid-base transporters can modify carcinogenesis and the behavior of cancer cells including cancer cell proliferation, migration, and invasion (*Boedtkjer and Pedersen, 2020*; *Amith and Fliegel, 2017*; *Stock and Pedersen, 2017*). Although detailed molecular mechanisms are not yet established, elevated $pH_i$ maintains metabolic activity (*Parks et al., 2013*), increases DNA and protein synthesis (*Pedersen, 2006*), and accelerates cell cycle progression (*Flinck et al., 2018*) in cultured cell lines. In accordance, mice with disrupted expression of NBCn1 show delayed tumor development and decelerated tumor growth when tested using models of carcinogen- and ErbB2-induced breast cancer (*Lee et al., 2016*; *Lee et al., 2018*).

Acidification at the outer cell surface depends on the rate of net acid transfer across the cell membrane and on diffusion hindrances that limit exchange with the bulk interstitial solution and the blood stream. Cell surface pH can modify cell-cell and cell-matrix interactions (*Stock et al., 2005*; *Riemann et al., 2019*), pH gradients from the leading to the rear end of cells can promote directional migration (*Stock et al., 2007*; *Boedtkjer et al., 2016*), and interstitial acidification of the tumor microenvironment has potential for modifying anti-cancer immune responses (*Cassim and Pouyssegur, 2019*). Indeed, cancer progression through metastasis and development of treatment resistance have been reported sensitive to inhibition of $Na^+/H^+$ exchangers (*Amith and Fliegel, 2017*; *Stock and Pedersen, 2017*); however, several anti-cancer effects of pharmacologically inhibiting $Na^+/H^+$ exchange appear only partly pH-dependent (*Boedtkjer et al., 2012*; *Loo et al., 2012*; *Schwab et al., 2012*) and can be caused by NHE1-independent toxicity due to intracellular drug accumulation (*Rolver et al., 2020*).

Despite the molecular and mechanistic insights from human cultured cell lines and inbred mouse models, the consequences of pH deregulation in human cancer tissue remain unclear. In particular, mechanisms of pH control in breast cancer tissue and their consequences for disease progression were previously explored in models that do not reflect the heterogeneity of human breast cancer. In the current study, we investigated an extensive cohort of human breast cancer patients, sufficiently large to reflect the heterogeneity of acid-base conditions and the variation in cellular handling of metabolic waste products. We tested the hypotheses that (a) specific clinical and pathological characteristics accelerate cellular net acid extrusion and determine the underlying molecular mechanisms of pH regulation in human breast cancer tissue; and (b) the capacity for cellular net acid extrusion,

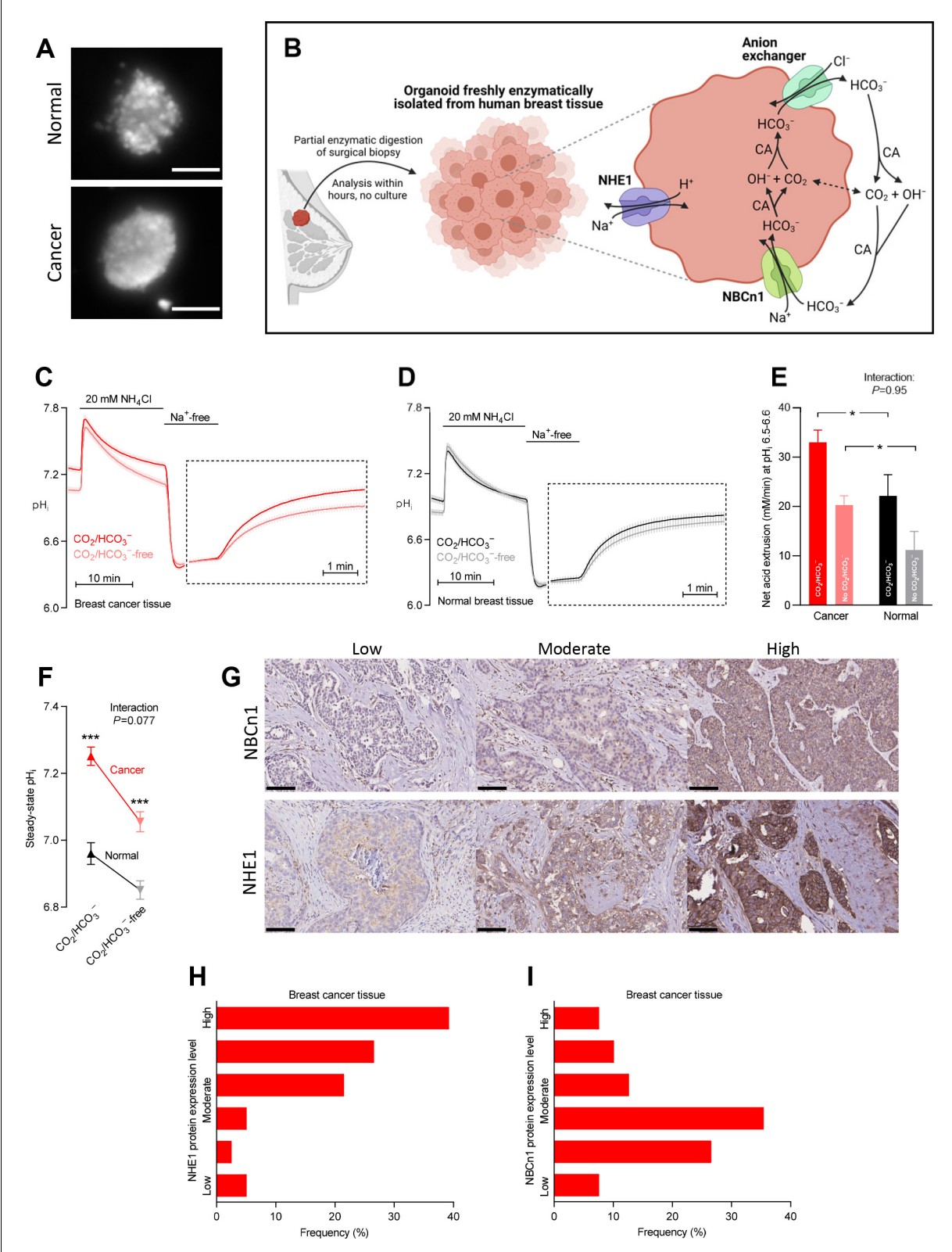

**Figure 1.** Cellular net acid extrusion in human breast cancer tissue and normal breast tissue relies on extracellular $Na^+$ and is partially $CO_2/HCO_3^-$-dependent consistent with the expression of $Na^+,HCO_3^-$ cotransporter NBCn1 and $Na^+/H^+$ exchanger NHE1. Moreover, steady-state intracellular pH ($pH_i$) and the capacity for net acid extrusion are elevated in human breast cancer tissue compared to normal breast tissue. (A) Exemplar fluorescence images of 2',7'-bis-(2-carboxyethyl)-5-(and-6)-carboxyfluorescein (BCECF)-loaded organoids freshly isolated from human breast cancer tissue (lower

*Figure 1 continued on next page*

*Figure 1 continued*

panel) and normal breast tissue (upper panel). The scale bars represent 100 µm. (B) Illustration of the experimental procedure for acute enzymatic isolation of organoids from human breast biopsies and a schematic showing the acid-base transporters involved in $pH_i$ control in breast cancer cells. The image was generated with Biorender.com. CA, carbonic anhydrase. (C,D) Traces of $NH_4^+$-prepulse-induced $pH_i$ dynamics in human breast cancer tissue (C, n=75–76) and normal breast tissue (D, n=48–49). The time scale within the dotted rectangles is expanded in order to improve resolution during the $pH_i$ recovery phase. (E) Cellular net acid extrusion activities in presence and nominal absence of $CO_2/HCO_3^-$ were calculated in the $pH_i$ range 6.5–6.6 for human breast cancer tissue and normal breast tissue (n=48–76). *Figure 1—figure supplement 1* provides a detailed analysis of the net acid extrusion capacity as function of $pH_i$. (F) Initial steady-state $pH_i$ in human breast cancer tissue (n=79–80) and normal breast tissue (n=49–50) in presence and nominal absence of $CO_2/HCO_3^-$. (G–I) Representative immunohistochemical images (G) and summarized pathologist-scored protein expression data for NHE1 (H, n=79) and NBCn1 (I, n=79) in human breast carcinomas. The size bars represent 100 µm. Data in panels E and F were compared by mixed-effects analyses followed by Sidak's multiple comparisons test. 'Interaction' reports whether the effect of $CO_2/HCO_3^-$ varies between breast cancer and normal breast tissue. *p<0.05, ***p<0.001 vs. normal breast tissue under similar conditions. *Figure 1—source data 1* contains the data pertaining to this figure and the de-identified clinicopathological information used to stratify data in *Figure 2, 3, 4, 5, 7* and *8* and *Figure 2—figure supplement 1*; this information is combined in the multiple linear and logistic regression analyses illustrated in *Figure 6*.

The online version of this article includes the following source data and figure supplement(s) for figure 1:

**Source data 1.** Data file containing NBCn1 and NHE1 protein expression levels, steady-state intracellular pH ($pH_i$) values, and net acid extrusion capacities linked to de-identified clinical and pathological patient characteristics.

**Figure supplement 1.** Cellular net acid extrusion activities in presence and nominal absence of $CO_2/HCO_3^-$ plotted as functions of intracellular pH ($pH_i$) for human breast cancer tissue and normal breast tissue.

**Figure supplement 1—source data 1.** Data file containing net acid extrusion capacities calculated at specified intracellular pH ($pH_i$) levels and linked to de-identified clinical and pathological patient characteristics.

the steady-state $pH_i$ level, and the expression of acid-base transporters in human breast cancer tissue predict the severity of disease.

## Results

We sampled human tissue biopsies from an extensive cohort of 110 women with breast cancer (*Table 1*) and evaluated $pH_i$ dynamics based on organoids freshly isolated from the breast cancer tissue and corresponding normal breast tissue (*Figure 1*). Within this patient population, we stratified the $pH_i$ dynamics and the NHE1 and NBCn1 expression levels by histopathology (*Figure 2* and *Figure 2—figure supplement 1*), malignancy grade (*Figure 3*), estrogen receptor status (*Figure 4*), and HER2 status (*Figure 5*); and adjusted for variation in other clinical and pathological characteristics (*Figure 6*). We then explored how the $pH_i$ dynamics and the NHE1 and NBCn1 protein expression levels relate to cancer cell proliferation (*Figure 7*) and lymph node metastasis (*Figure 8*). Finally, we evaluated how variation in expression levels for acid-base transporters influence patient survival within individual breast cancer molecular subtypes (*Figures 9* and *10*, and *Figure 10—figure supplements 1* and *2*).

### The elevated net acid extrusion capacity is Na⁺- and partly $CO_2/HCO_3^-$-dependent in human breast cancer tissue

We freshly isolated organoids from human breast biopsies by partial collagenase digestion (*Figure 1A,B*). We prepared the organoids in immediate continuation of the breast-conserving surgery and investigated them directly after isolation without culture in order to avoid phenotypic changes. We previously confirmed that organoids freshly isolated from breast tissue biopsies consist predominantly of cytokeratin-19-positive epithelial cells with few smooth muscle α-actin-positive myofibroblasts (*Lee et al., 2016*; *Lee et al., 2015*).

We experimentally induced intracellular acidification by $NH_4^+$-prepulse technique (*Boron and De Weer, 1976*; *Boedtkjer and Aalkjaer, 2012*) as illustrated in *Figure 1C* for breast cancer tissue and *Figure 1D* for normal breast tissue. Addition of $NH_4Cl$ to the experimental bath solution acutely elevates $pH_i$ as $NH_3$ traverses plasma membranes and binds $H^+$ from the cytosol. The subsequent gradual decline of $pH_i$ occurs when $NH_4^+$ enters cells—predominantly through plasma membrane $K^+$ conductances (e.g., $K^+$ channels and $Na^+/K^+$-ATPases)—and base equivalents are extruded, for instance, through $Cl^-/HCO_3^-$ exchange. Washout of $NH_4Cl$ then causes $NH_3$ to rapidly leave the cells; and intracellular acidification ensues when $H^+$, as a consequence, is liberated from intracellular $NH_4^+$ and accumulates in the cytosol.

**Table 1.** Clinical and pathological characteristics of the patient cohort investigated for intracellular pH ($pH_i$) dynamics and protein expression.

| | |
|---|---|
| Number of patients | 110 |
| Patient age (years; median, interquartile range) | 64.5 (56–74) |
| Tumor size (mm; median, interquartile range) | 18 (14–26) |
| Histological type | |
| Invasive ductal carcinomas | 92 (84%) |
| Invasive lobular carcinomas | 10 (9%) |
| Mucinous adenocarcinomas | 5 (5%) |
| Tubular carcinoma | 2 (2%) |
| Pleomorphic lobular carcinoma | 1 (1%) |
| HER2 receptor status | |
| Normal | 95 (86%) |
| Overexpression or gene amplification | 15 (14%) |
| Estrogen receptor status | |
| 90–100% $ER^+$ cells | 99 (90%) |
| 0–15% $ER^+$ cells | 11 (10%) |
| Malignancy grade | |
| I | 31 (28%) |
| II | 52 (47%) |
| III | 22 (20%) |
| Not graded | 5 (5%) |
| Axillary lymph node status | |
| Negative | 69 (63%) |
| Isolated tumor cells | 15 (14%) |
| Micro-metastases | 5 (5%) |
| Macro-metastases | 21 (19%) |
| Ki67 index | |
| 0–30% $Ki67^+$ cells | 82 (75%) |
| 35–90% $Ki67^+$ cells | 28 (25%) |

Assessing the patient population as a whole, we observed that cellular net acid extrusion during intracellular acidification was almost fully dependent on extracellular $Na^+$ and relied on both $CO_2$/$HCO_3^-$-dependent and -independent transport mechanisms (*Figure 1C,D* and *Figure 1—figure supplement 1*). As schematically illustrated in *Figure 1B*, these observations support previous reports (*Boedtkjer et al., 2013*; *Lee et al., 2015*) that $Na^+,HCO_3^-$ cotransporters and $Na^+/H^+$ exchangers are mainly responsible for cellular net acid extrusion in human breast cancer tissue. We observed upregulated capacity for $Na^+/H^+$ exchange activity during carcinogenesis based on the faster $Na^+$-dependent $pH_i$ recovery in organoids freshly isolated from breast cancer tissue compared to normal breast tissue when evaluated in the nominal absence of $CO_2$/$HCO_3^-$ (*Figure 1C,D*). Likewise, we detected contribution from $Na^+,HCO_3^-$ cotransport as the ability to recover $pH_i$ faster and at more alkaline $pH_i$ when $CO_2$/$HCO_3^-$ was present (*Figure 1C,D*).

The $pH_i$ traces (*Figure 1C,D*) recorded from freshly processed breast tissue biopsies allowed us to evaluate (a) the initial steady-state $pH_i$ level with extracted values summarized in *Figure 1F* and (b) the capacity for cellular net acid extrusion during $pH_i$ recovery from $NH_4^+$-prepulse-induced acidification (dotted rectangles in *Figure 1C,D*) with calculated values summarized in *Figure 1E*. As the acid extrusion mechanisms activate at low $pH_i$ (*Figure 1—figure supplement 1*), their activities must be compared at equivalent $pH_i$ values: at $pH_i$ 6.5–6.6, we demonstrated a greater capacity for net acid extrusion in human breast cancer tissue compared to normal breast tissue, whether examined in

the presence or nominal absence of $CO_2/HCO_3^-$ (*Figure 1E*). The net acid extrusion capacity in breast cancer tissue was upregulated predominantly in the near-neutral $pH_i$ range (compare *Figure 1—figure supplement 1A and B*), which was also reflected in an elevated steady-state $pH_i$ when human breast cancer tissue was investigated under similar experimental conditions as normal breast tissue either with or without $CO_2/HCO_3^-$ present (*Figure 1F*). The drop in steady-state $pH_i$ in response to nominal omission of $CO_2/HCO_3^-$ (*Figure 1F*) supports a greater contribution from $Na^+$, $HCO_3^-$ cotransport compared to anion exchange in the near-physiological $pH_i$ range (*Figure 1B*).

In congruence with the functional observations, we identified prominent protein expression of NHE1 and NBCn1 in the human breast cancer tissue and considerable inter-individual variation within the evaluated patient population (*Figure 1G–I*).

## $Na^+$,$HCO_3^-$ cotransport is more pronounced in invasive lobular than ductal breast carcinomas

Breast cancer is histopathologically diverse, but invasive ductal and lobular breast carcinomas are most frequent (*Table 1*). We evaluated whether the mechanisms of net acid extrusion differ between tumors of separate histopathologies (*Figure 2* and *Figure 2—figure supplement 1*). Invasive lobular (*Figure 2A* and *Figure 2—figure supplement 2A*) and ductal (*Figure 2B* and *Figure 2—figure supplement 2B*) breast carcinomas both showed dual dependency on $Na^+/H^+$ exchange and $Na^+$, $HCO_3^-$ cotransport for $pH_i$ regulation. We observed a tendency toward greater relative capacity for net acid extrusion via $Na^+$,$HCO_3^-$ cotransport during intracellular acidification in invasive lobular than ductal breast carcinomas (*Figure 2C*). This increased influence of $Na^+$,$HCO_3^-$ cotransport in invasive lobular carcinomas was more pronounced with respect to steady-state $pH_i$ control where it reached statistical significance both before (*Figure 2D*) and after (*Figure 6B*) we adjusted for variation in other clinicopathological characteristics (i.e., patient age, tumor size, malignancy grade, estrogen receptor expression, and HER2 status).

Expression of NHE1 and NBCn1 protein was generally moderate in invasive lobular carcinomas and showed less inter-individual variation than observed for invasive ductal carcinomas (*Figures 2E, F* and *6E,F*).

Although our functional data cover too few patients with mucinous adenocarcinomas (n=5) to perform a formal comparison with the other histopathologies, the pattern of $pH_i$ regulation (*Figure 2—figure supplement 1A*) was similar to that observed in invasive lobular and ductal breast carcinomas (*Figure 2A,B*). Notably, the net acid extrusion capacity (*Figure 2—figure supplement 1B, C*) and steady-state $pH_i$ (*Figure 2—figure supplement 1D*) in mucinous adenocarcinomas confirmed the dual dependency on $Na^+/H^+$ exchange and $Na^+$,$HCO_3^-$ cotransport.

## $Na^+$,$HCO_3^-$ cotransport elevates $pH_i$ more in breast carcinomas of high malignancy grade

Malignancy grading of breast cancer tissue provides valuable prognostic information (*Rakha et al., 2010*), and we show here that the malignancy grade is also reflected in the $pH_i$ dynamics of the breast cancer tissue (*Figure 3A–C*). We observed a tendency toward a greater net acid extrusion capacity in higher malignancy grade breast carcinomas (*Figure 3D*, *Figure 3—figure supplement 1*, and *Figure 6D*). More prominently, we found that the $Na^+$,$HCO_3^-$ cotransport activity established an increasingly elevated steady-state $pH_i$ in breast carcinomas with higher malignancy grade, and this effect reached statistical significance both before (*Figure 3E*) and after (*Figure 6B*) adjustment for other clinicopathological characteristics.

Protein expression levels for NHE1 and NBCn1 did not significantly differ between breast cancer tissue of different malignancy grades (*Figures 3F,G* and *6E,F*).

## $Na^+/H^+$ exchange elevates $pH_i$ more in breast carcinomas with low estrogen receptor expression

Estrogen receptors play important roles in mammary gland development (*Shyamala, 1997*). The expression of estrogen receptors in breast cancer tissue varied considerably between patients: as shown in *Table 1*, there was a clear distinction between a large group of patients with tumors showing widespread estrogen receptor expression (≥90% positive cells) and a smaller subset of patients with tumors showing very limited estrogen receptor expression (≤15% positive cells). Previous

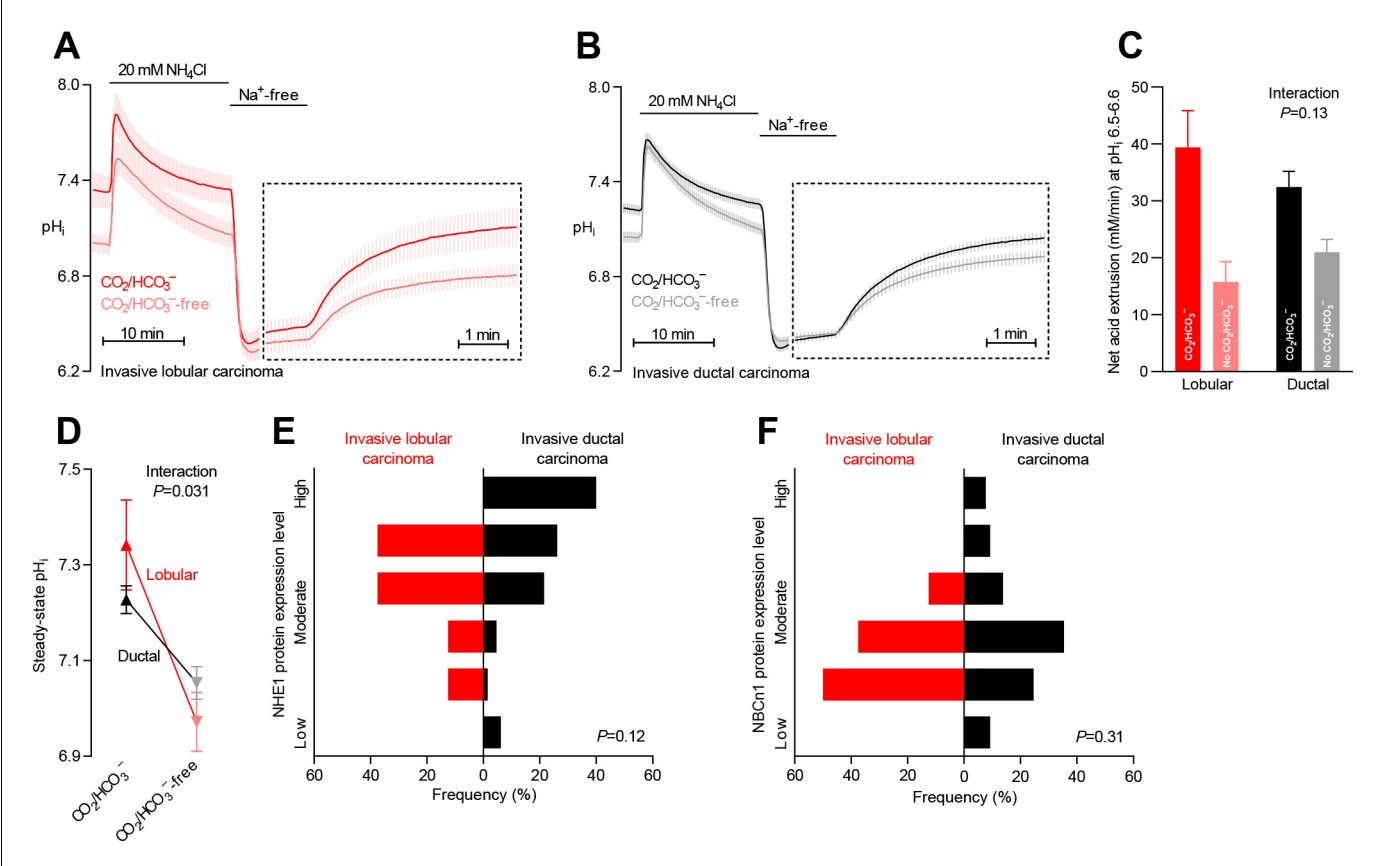

**Figure 2.** $Na^+,HCO_3^-$ cotransport is critical for cellular net acid extrusion and steady-state intracellular pH ($pH_i$) in human invasive ductal and, particularly, lobular breast carcinomas. (**A,B**) Traces of $NH_4^+$-prepulse-induced $pH_i$ dynamics in human invasive lobular (**A**, n=8–9) and ductal (**B**, n=60–62) breast carcinomas. The time scale within the dotted rectangles is expanded in order to improve resolution during the $pH_i$ recovery phase. (**C**) Cellular net acid extrusion activities in presence and nominal absence of $CO_2/HCO_3^-$ were calculated in the $pH_i$ range 6.5–6.6 for human invasive lobular and ductal breast carcinomas (n=8–62). *Figure 2—figure supplement 2* provides a detailed analysis of the net acid extrusion capacity as function of $pH_i$. (**D**) Initial steady-state $pH_i$ in human invasive lobular (n=9) and ductal (n=64–65) breast carcinomas in presence and nominal absence of $CO_2/HCO_3^-$. Data in panels C and D were compared by mixed-effects analyses. 'Interaction' reports whether the effect of $CO_2/HCO_3^-$ varies between human invasive lobular and ductal breast carcinomas. (**E,F**) Summarized pathologist-scored, immunohistochemistry-based protein expression data for NHE1 (**E**) and NBCn1 (**F**) in human invasive lobular (n=8) and ductal (n=65) breast carcinomas. Protein expression in human invasive lobular and ductal breast carcinomas was compared by $\chi^2$ tests for trend. *Figure 2—figure supplement 1* provides data from mucinous adenocarcinomas.

The online version of this article includes the following figure supplement(s) for figure 2:

**Figure supplement 1.** $Na^+/H^+$ exchange and $Na^+,HCO_3^-$ cotransport activity regulate intracellular pH ($pH_i$) in mucinous adenocarcinomas.

**Figure supplement 2.** Cellular net acid extrusion activities in presence and nominal absence of $CO_2/HCO_3^-$ plotted as functions of intracellular pH ($pH_i$) for human invasive lobular and ductal breast carcinomas.

studies have found that estrogen receptors are expressed in around 10% of the cells in normal breast epithelium (*Oh et al., 2017*).

We detected no obvious effect of estrogen receptor expression on the net acid extrusion capacity of human breast cancer tissue during intracellular acidification (*Figure 4A–C*, *Figure 4—figure supplement 1*, and *Figure 6C,D*). However, as illustrated in *Figure 4D*, we observed elevated steady-state $pH_i$ in breast cancer tissue with no or very low expression (0–15% $ER^+$) compared to breast cancer tissue with high expression (90–100% $ER^+$) of estrogen receptors. This effect was explained by a greater contribution of $Na^+/H^+$ exchange activity—as it was evident both in the presence and absence of $CO_2/HCO_3^-$—and became statistically significant after adjustment for other clinicopathological characteristics (*Figure 6A*).

The protein expression levels for NHE1 and NBCn1 were not significantly influenced by estrogen receptor status (*Figures 4E,F* and *6E,F*).

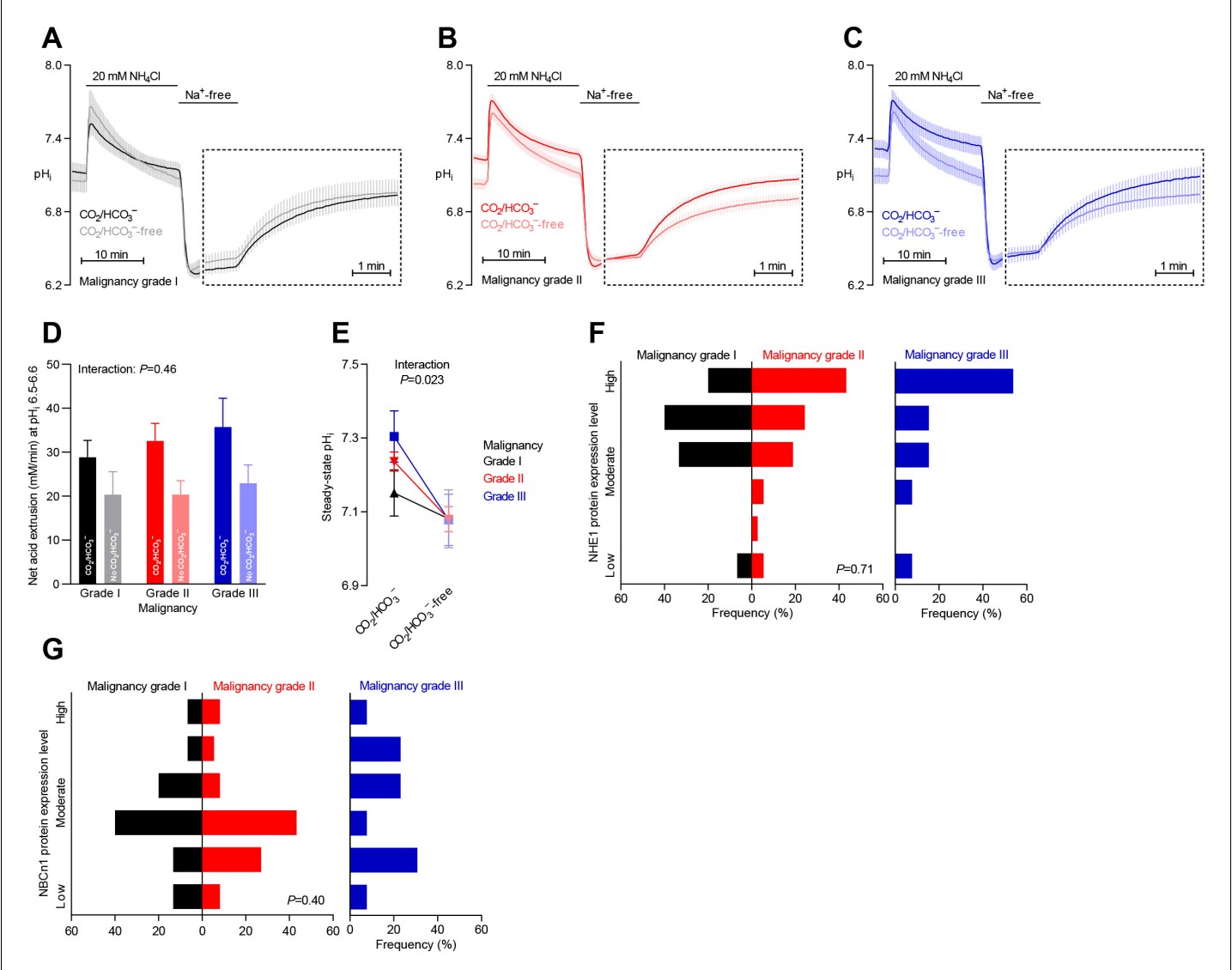

**Figure 3.** Steady-state intracellular pH ($pH_i$) is elevated in human breast carcinomas of high malignancy grade due to cellular $CO_2/HCO_3^-$-dependent net acid extrusion. (**A–C**) Traces of $NH_4^+$-prepulse-induced $pH_i$ dynamics in human invasive ductal breast carcinomas of malignancy grades I (**A**, n=12–14), II (**B**, n=34), and III (**C**, n=14). The time scale within the dotted rectangles is expanded in order to improve resolution during the $pH_i$ recovery phase. (**D**) Cellular net acid extrusion activities in presence and nominal absence of $CO_2/HCO_3^-$ were calculated in the $pH_i$ range 6.5–6.6 for human invasive ductal breast carcinomas of malignancy grades I, II, and III (n=12–34). *Figure 3—figure supplement 1* provides a detailed analysis of the net acid extrusion capacity as function of $pH_i$. (**E**) Initial steady-state $pH_i$ in human invasive ductal breast carcinomas of malignancy grades I (n=14–15), II (n=35), and III (n=15) in presence and nominal absence of $CO_2/HCO_3^-$. Data in panels D and E were compared by mixed-effects analyses and repeated-measures one-way ANOVA with post-test for linear trend. 'Interaction' reports whether the effect of $CO_2/HCO_3^-$ varies between human breast carcinomas of malignancy grades I, II, and III. (**F, G**) Summarized pathologist-scored, immunohistochemistry-based protein expression data for NHE1 (**F**, n=65) and NBCn1 (**G**, n=65) in human invasive ductal carcinomas stratified by malignancy grade. Protein expression in human breast carcinomas of malignancy grades I, II, and III was compared by $\chi^2$ tests.

The online version of this article includes the following figure supplement(s) for figure 3:

**Figure supplement 1.** Cellular net acid extrusion activities in presence and nominal absence of $CO_2/HCO_3^-$ plotted as functions of intracellular pH ($pH_i$) for human invasive ductal breast carcinomas of malignancy grades I, II, and III.

## HER2 receptors increase expression of NBCn1 and NHE1 protein

Growth factor input facilitates cancer cell proliferation and the malignant phenotype of cancer cells; and the functional implications of HER2 receptors are amplified by overexpression or gene

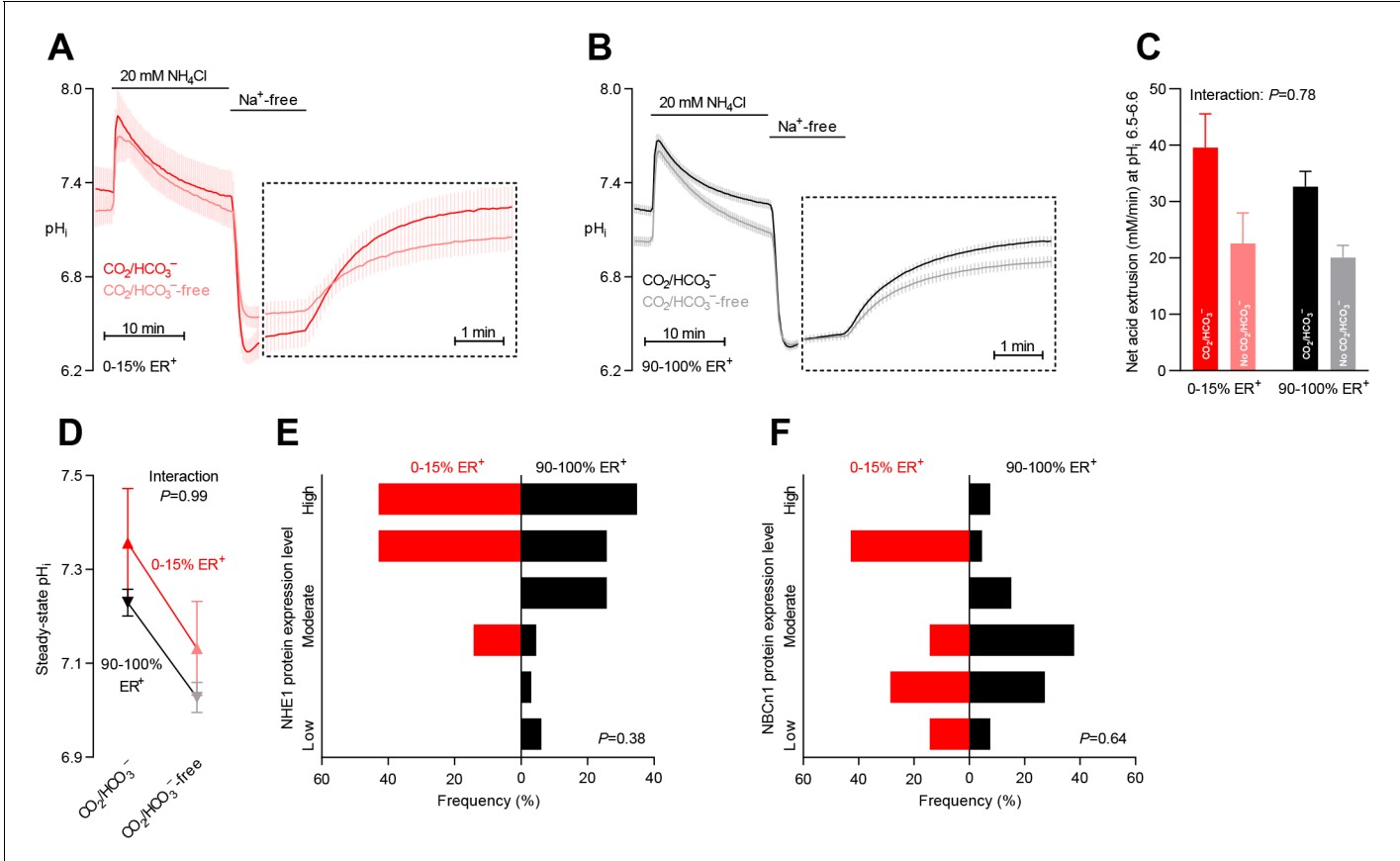

**Figure 4.** Intracellular pH ($pH_i$) is elevated in estrogen receptor-negative breast cancer. (**A,B**) Traces of $NH_4^+$-prepulse-induced $pH_i$ dynamics in human breast carcinomas stratified by estrogen receptor status (**A**: negative, n=6; **B**: positive, n=63–64). The time scale within the dotted rectangles is expanded in order to improve resolution during the $pH_i$ recovery phase. (**C**) Cellular net acid extrusion activities in presence and nominal absence of $CO_2/HCO_3^-$ were calculated in the $pH_i$ range 6.5–6.6 for human breast carcinomas stratified by estrogen receptor status (n=6–64). *Figure 4—figure supplement 1* provides a detailed analysis of the net acid extrusion capacity as function of $pH_i$. (**D**) Initial steady-state $pH_i$ in human estrogen receptor-negative (0–15% $ER^+$, n=7–8) and -positive (90–100% $ER^+$, n=66–67) breast carcinomas. Data in panels C and D were compared by mixed-effects analyses. 'Interaction' reports whether the effect of $CO_2/HCO_3^-$ varies between estrogen receptor-negative and -positive breast carcinomas. (**E,F**) Pathologist-scored, immunohistochemistry-based protein expression levels for NHE1 (**E**) and NBCn1 (**F**) in human breast carcinomas (n=73) stratified by estrogen receptor status. Protein expression in human estrogen receptor-negative and -positive breast carcinomas was compared by $\chi^2$ tests for trend. The online version of this article includes the following figure supplement(s) for figure 4:

**Figure supplement 1.** Cellular net acid extrusion activities in presence and nominal absence of $CO_2/HCO_3^-$ plotted as function of intracellular pH ($pH_i$) for human breast carcinomas stratified by estrogen receptor status.

amplification in 10–20% of breast cancer patients (*Table 1*) and less commonly by activating somatic mutations (*Connell and Doherty, 2017*).

HER2 receptor status did not significantly influence the net acid extrusion capacity of breast cancer tissue during intracellular acidification (*Figure 5A–C* and *Figure 5—figure supplement 1*). Whereas we observed a strong tendency toward a higher $CO_2/HCO_3^-$-dependent rise in steady-state $pH_i$ in HER2-positive tumors (*Figure 5D*), this effect was substantially attenuated when adjusted for other clinicopathological characteristics (*Figure 6B*).

The protein expression levels of NHE1 as well as NBCn1 were elevated in breast carcinomas with HER2 overexpression or gene amplification both before (*Figure 5E,F*) and after (*Figure 6E,F*) adjustment for other clinicopathological characteristics.

## Patient age and tumor size

We next plotted steady-state $pH_i$ levels (*Figure 6—figure supplement 1A,B*) and capacities for net acid extrusion during intracellular acidification (*Figure 6—figure supplement 1C,D*) as functions of

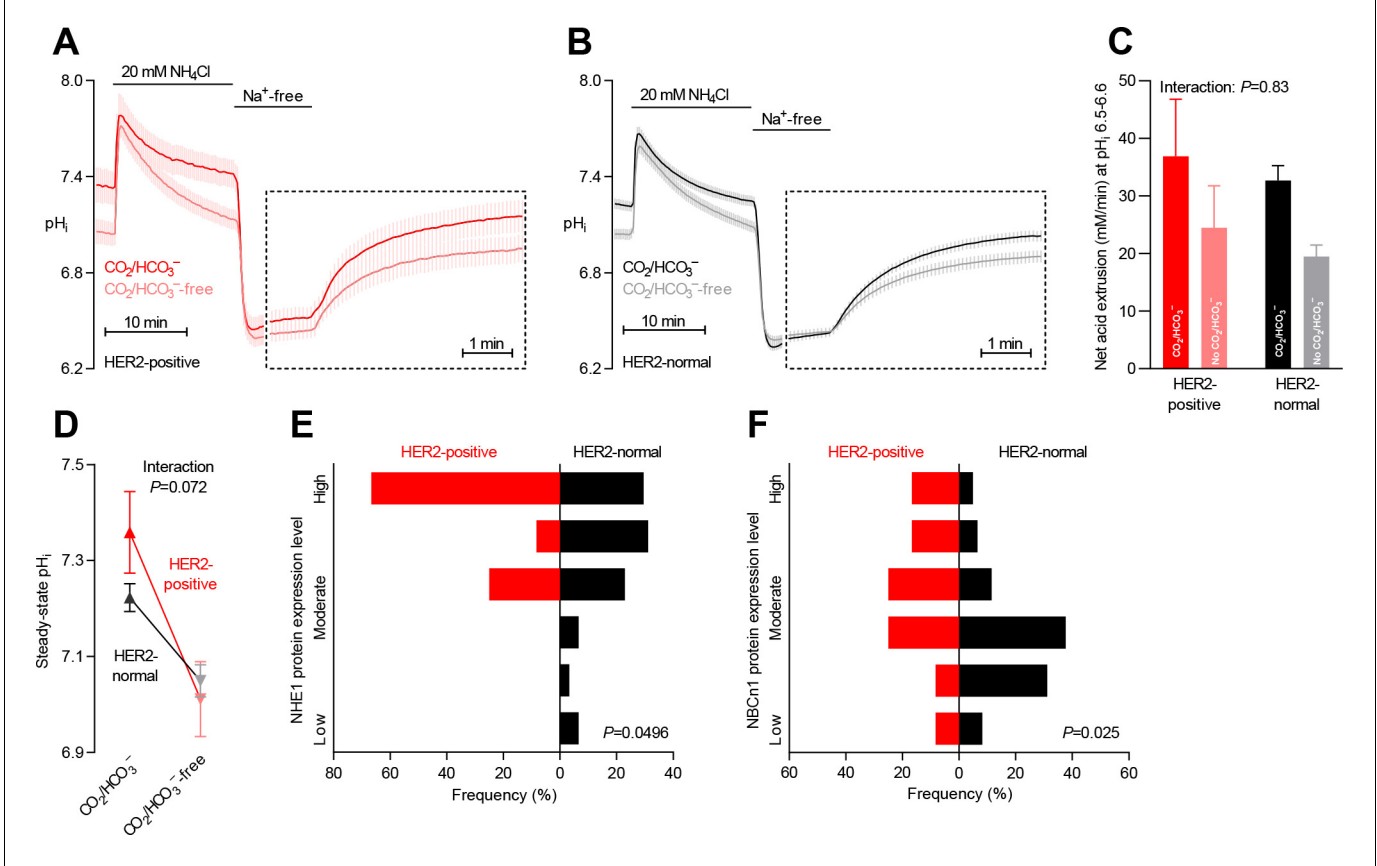

**Figure 5.** Protein expression of NHE1 and NBCn1 is elevated in HER2-positive human breast cancer. (A,B) Traces of $NH_4^+$-prepulse-induced intracellular pH ($pH_i$) dynamics in human HER2-positive (A, n=9–11) and HER2-normal (B, n=58–61) breast carcinomas. The time scale within the dotted rectangles is expanded in order to improve resolution during the $pH_i$ recovery phase. (C) Cellular net acid extrusion activities in the presence and nominal absence of $CO_2/HCO_3^-$ were calculated in the $pH_i$ range 6.5–6.6 for human breast carcinomas stratified by HER2 status (n=9–61). ***Figure 5—figure supplement 1*** provides a detailed analysis of the net acid extrusion capacity as function of $pH_i$. (D) Initial steady-state $pH_i$ in HER2-positive (n=10–11) and -normal (n=63) human breast carcinomas. Data in panels C and D were compared by mixed-effects analyses. 'Interaction' reports whether the effect of $CO_2/HCO_3^-$ varies between HER2-positive and -normal breast carcinomas. (E,F) Summarized pathologist-scored, immunohistochemistry-based protein expression data for NHE1 (E) and NBCn1 (F) in human breast carcinomas (n=73) stratified by HER2 status. Protein expression in human HER2-positive and -normal breast carcinomas was compared by $\chi^2$ tests for trend.

The online version of this article includes the following figure supplement(s) for figure 5:

**Figure supplement 1.** Cellular net acid extrusion activities in the presence and nominal absence of $CO_2/HCO_3^-$ plotted as functions of intracellular pH ($pH_i$) for human breast carcinomas stratified by HER2 status.

tumor size (***Figure 6—figure supplement 1A,C***) and patient age (***Figure 6—figure supplement 1B, D***). The $CO_2/HCO_3^-$-independent net acid extrusion capacity decreased as function of patient age before (***Figure 6—figure supplement 1D***) but not after (***Figure 6C***) adjustment for other clinicopathological characteristics, whereas none of the other plots revealed significant correlations (***Figure 6A–D*** and ***Figure 6—figure supplement 1A–D***).

## Clinicopathological characteristics independently predict acid-base dynamics and expression of acid-base transporters

The patient groups illustrated in ***Figures 2–5*** are stratified by individual clinicopathological parameters. We next performed multiple linear and logistic regression analyses to control for multiple comparisons, take into account unbalanced distributions within individual groups, and identify clinical and pathological characteristics that independently predict steady-state $pH_i$ (***Figure 6A,B***), net acid extrusion capacity (***Figure 6C,D***), or acid-base transporter expression (***Figure 6E,F***).

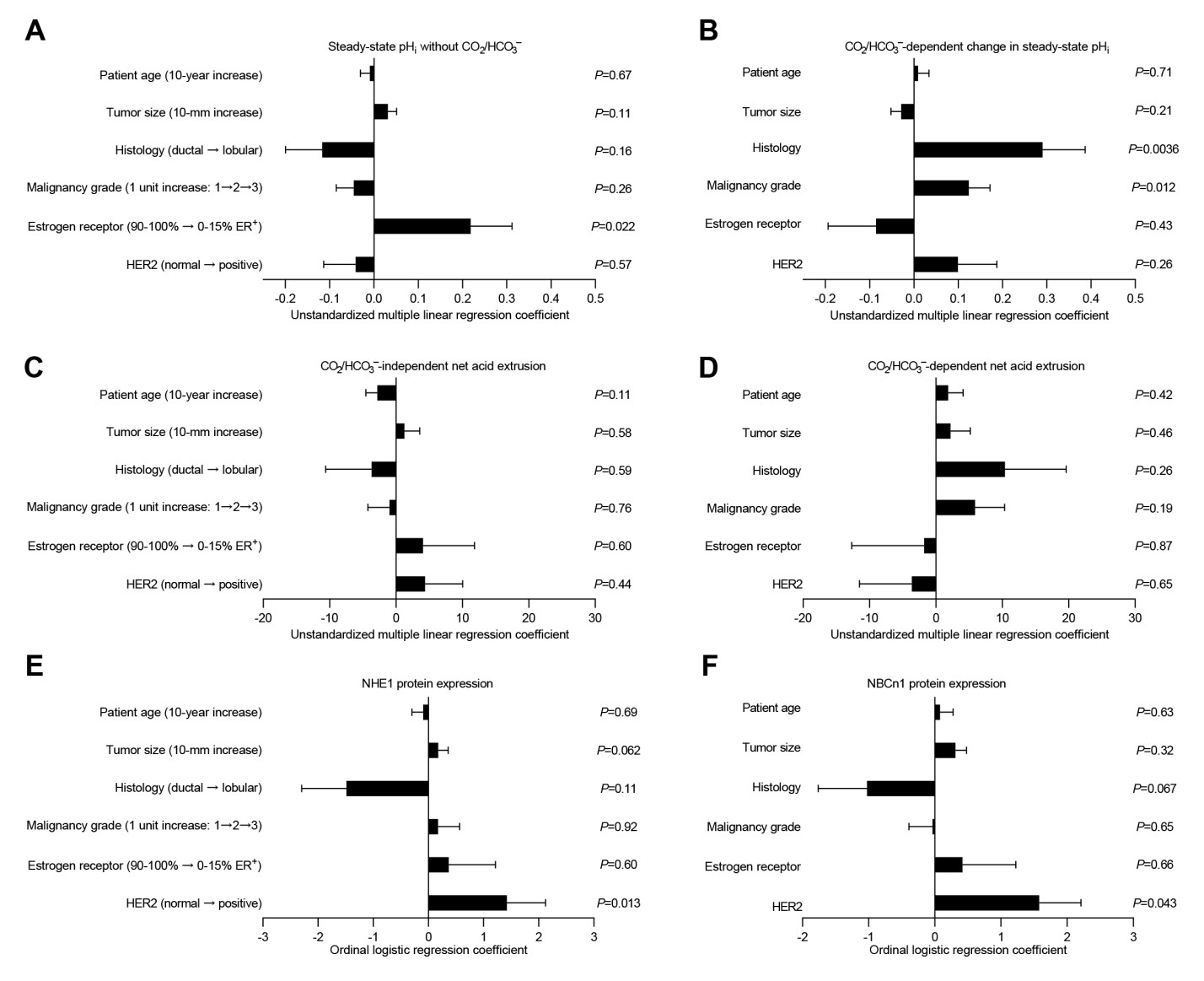

**Figure 6.** Histology, malignancy grade, and receptor expression profiles are independent predictors of intracellular pH (pH$_i$) dynamics and acid-base transporter expression in human invasive lobular and ductal breast carcinomas. (**A–D**) Multiple linear regression analyses show the independent influences of patient age, tumor size, histology, malignancy grade, and expression of estrogen and HER2 receptors on steady-state pH$_i$ and net acid extrusion capacity in human breast cancer tissue (n=78). Data were adjusted for inter-investigator variation. ***Figure 6—figure supplement 1*** provides plots of steady-state pH$_i$ and net acid extrusion as functions of tumor size and patient age. (**E,F**) Ordinal logistic regression analyses show the independent influences of patient age, tumor size, histology, malignancy grade, and expression of estrogen and HER2 receptors on protein expression levels for NHE1 (**E**) and NBCn1 (**F**) in human breast cancer tissue (n=73).

The online version of this article includes the following figure supplement(s) for figure 6:

**Figure supplement 1.** Plots of acid-base parameters in human breast cancer tissue (n=64–74) as functions of tumor size (**A,C**) and patient age (**B,D**).

Estrogen receptor status was the predominant influence on steady-state pH$_i$ in absence of $CO_2$/$HCO_3^-$ (***Figure 6A***), whereas histology and malignancy grade were the predominant modifiers of the $CO_2$/$HCO_3^-$-dependent increase in steady-state pH$_i$ (***Figure 6B***).

None of the tested clinical or pathological parameters (i.e., patient age, tumor size, histology, malignancy grade, estrogen receptor status, HER2 status) showed independent value to predict the capacity for $CO_2$/$HCO_3^-$-independent net acid extrusion mediated by Na$^+$/H$^+$ exchange (***Figure 6C***)

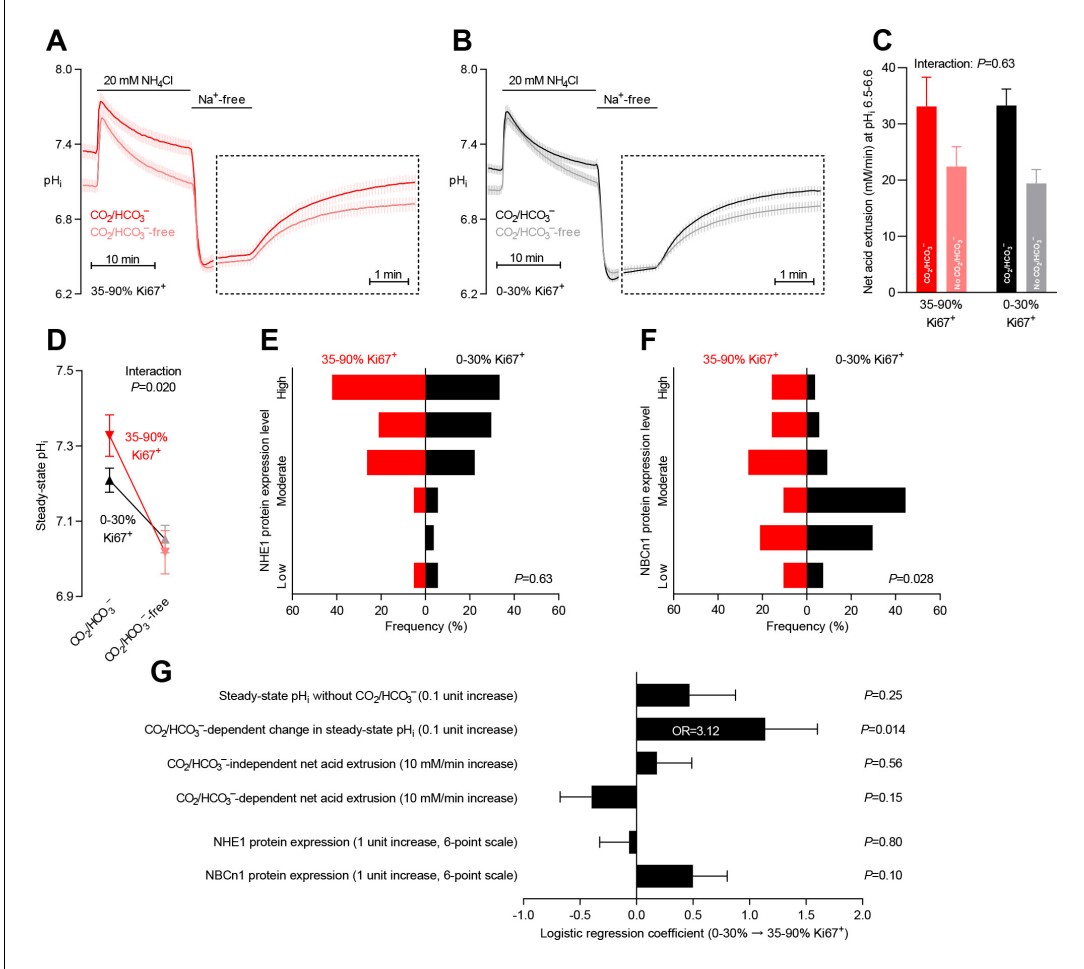

**Figure 7.** Steady-state intracellular pH (pH$_i$) is elevated in human breast carcinomas with high proliferative activity (elevated Ki67 index). (A,B) Traces of NH$_4^+$-prepulse-induced pH$_i$ dynamics in breast carcinomas with high (A, 35–90% Ki67$^+$, n=19–20) and low (B, 0–30% Ki67$^+$, n=49–51) Ki67 index. The time scale within the dotted rectangles is expanded in order to improve resolution during the pH$_i$ recovery phase. (C) Cellular net acid extrusion activities in presence and nominal absence of CO$_2$/HCO$_3^-$ were calculated in the pH$_i$ range 6.5–6.6 for human breast carcinomas stratified by Ki67 index (n=19–51). *Figure 7—figure supplement 1* provides a detailed analysis of the net acid extrusion capacity as function of pH$_i$. (D) Initial steady-state pH$_i$ in breast carcinomas with high (n=20) and low (n=53–54) Ki67 index. Data in panels C and D were compared by mixed-effects analyses. 'Interaction' reports whether the effect of CO$_2$/HCO$_3^-$ varies between breast carcinomas with high and low Ki67 index. (E,F) Summarized pathologist-scored, immunohistochemistry-based protein expression levels for NHE1 (E) and NBCn1 (F) in human breast carcinomas (n=73) stratified by Ki67 index. Protein expression in human breast carcinomas of low and high Ki67 index was compared by $\chi^2$ tests for trend. (G) Results of binominal logistic regression analyses where the influence of the acid-base parameters and transporter expression levels on cellular proliferation was adjusted for patient age, tumor size, histology, malignancy grade, estrogen receptor status, HER2 status, and inter-investigator variation (n=73–78). Odds ratios (OR) are given for variables showing statistically significant association.

The online version of this article includes the following figure supplement(s) for figure 7:

**Figure supplement 1.** Cellular net acid extrusion activities in presence and nominal absence of CO$_2$/HCO$_3^-$ plotted as functions of intracellular pH (pH$_i$) for human breast carcinomas stratified by Ki67 index.

or the capacity for CO$_2$/HCO$_3^-$-dependent net acid extrusion mediated by Na$^+$,HCO$_3^-$ cotransport (*Figure 6D*) during intracellular acidification.

The NHE1 and NBCn1 protein expression levels in the human breast cancer tissue were independently elevated by HER2 overexpression or gene amplification (*Figure 6E,F*).

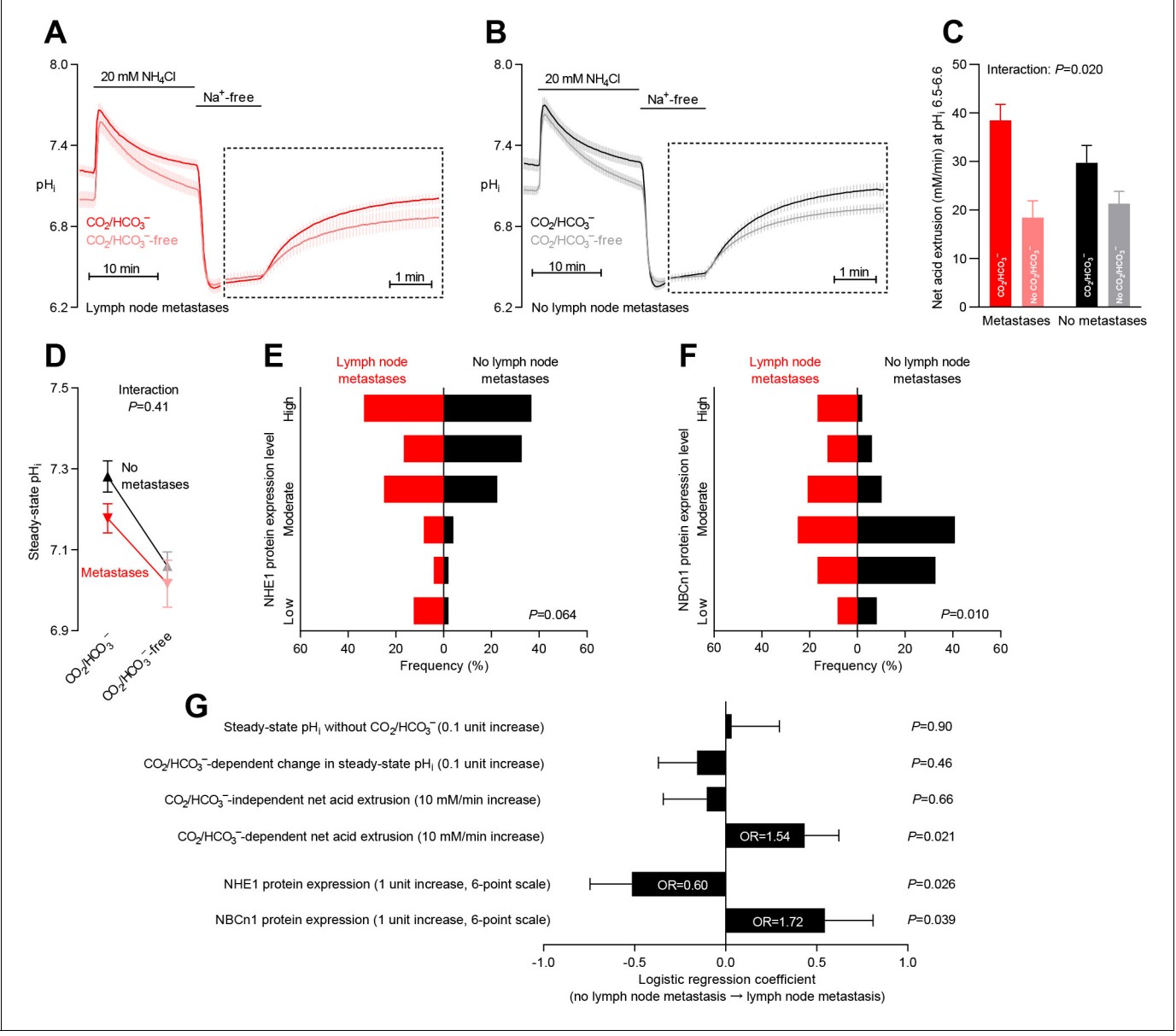

**Figure 8.** Primary breast cancer tissue from patients with axillary lymph node metastases shows higher $Na^+,HCO_3^-$ cotransport activity during intracellular acidification, higher NBCn1 expression, and lower NHE1 expression than breast cancer tissue from patients without metastases. (A,B) Traces of $NH_4^+$-prepulse-induced intracellular pH ($pH_i$) dynamics in primary breast carcinomas from patients with (A, n=25–28) and without (B, n=42–44) axillary lymph node metastases. Time scales within the dotted rectangles are expanded to improve resolution during the $pH_i$ recovery. (C) Cellular net acid extrusion activities in presence and absence of $CO_2/HCO_3^-$ calculated in the $pH_i$ range 6.5–6.6 for primary breast carcinomas stratified by axillary lymph node status (n=25–44). *Figure 8—figure supplement 1* provides detailed analysis of net acid extrusion capacities as function of $pH_i$. (D) Initial steady-state $pH_i$ in primary breast carcinomas from patients with (n=28) and without (n=45–46) lymph node metastases. Data in panels C and D were compared by mixed-effects analyses. 'Interaction' reports whether the effect of $CO_2/HCO_3^-$ varies between breast carcinomas from patients with and without lymph node metastases. (E,F) Summarized pathologist-scored, immunohistochemistry-based protein expression levels for NHE1 (E) and NBCn1 (F) in primary breast carcinomas from patients with (n=24) and without (n=49) lymph node metastases. Protein expression data were compared by $\chi^2$ tests for trend. (G) Results of binominal logistic regression analyses where influences of acid-base parameters and transporter expression levels on lymph node metastasis were adjusted for patient age, tumor size, histology, malignancy grade, estrogen receptor status, HER2 status, and inter-investigator variation (n=73–78). Odds ratios (OR) are given for variables showing statistically significant association.

The online version of this article includes the following figure supplement(s) for figure 8:

**Figure supplement 1.** Cellular net acid extrusion activities in presence and nominal absence of $CO_2/HCO_3^-$ plotted as functions of intracellular pH ($pH_i$) for human primary breast carcinomas stratified by axillary lymph node status.

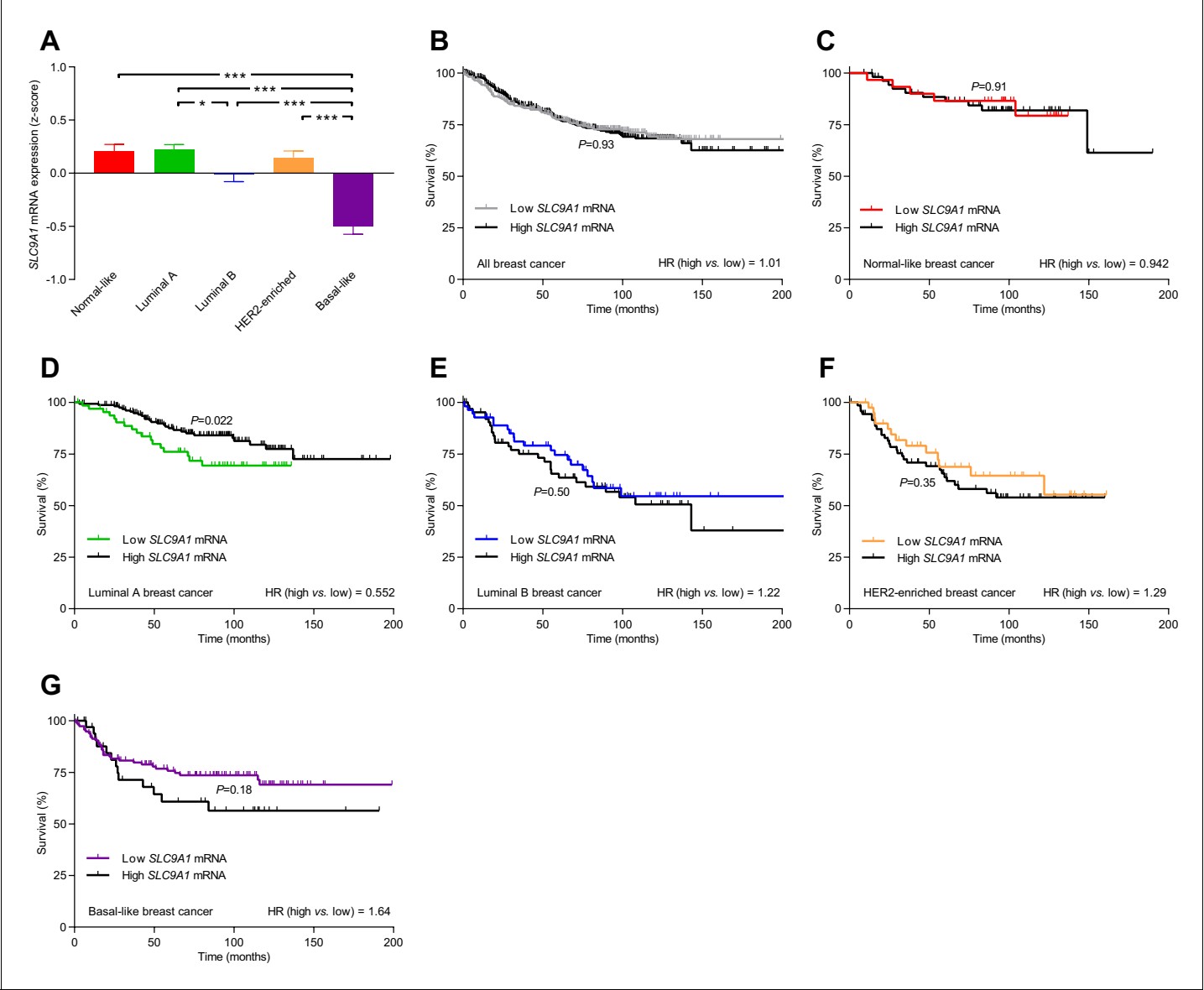

**Figure 9.** The levels of *SLC9A1* mRNA, encoding NHE1, vary among breast cancer subtypes, and high *SLC9A1* expression is associated with improved survival in patients with luminal A breast cancer. (**A**) Variation in *SLC9A1* mRNA levels among patients with different breast cancer subtypes (n=135–344). Expression data were compared by one-way ANOVA followed by Tukey's post-test. *Figure 9—figure supplement 1* provides data on mRNA expression of *ESR1*, *PGR*, *ERBB2*, *MKI67*, *LDHA*, *PECAM1*, and *CD34* in breast cancer tissue of the different molecular subtypes. *Figure 9—source data 1* provides results from correlation analyses between *SLC9A1* mRNA expression and the mRNA levels for *ESR1*, *PGR*, *ERBB2*, *SLC4A7*, *SLC16A1*, and *SLC16A3*. *p<0.05, ***p<0.001. (**B–G**) Survival curves stratified by *SLC9A1* mRNA levels in patients with different breast cancer subtypes. The ticks on the curves represent censored subjects. Survival data were compared by Mantel-Cox and Gehan-Breslow-Wilcoxon tests. HR, hazard ratio.

The online version of this article includes the following source data and figure supplement(s) for figure 9:

**Source data 1.** The mRNA expression for *SLC9A1* correlates with that of *ESR1* and *ERBB2*.

**Figure supplement 1.** Transcript patterns for the different molecular subtypes of breast cancer.

## Elevated pH$_i$ predicts high proliferative activity

The accentuated metabolism of cancer cells supplies chemical intermediates and energy for cell proliferation. However, the accelerated metabolism also leads to a higher cellular acid load and risk of intracellular acidification, which can limit further cell proliferation (*Boedtkjer and Pedersen, 2020*).

When we stratified the patient cohort by expression of proliferative markers (*Figure 7A–D* and *Figure 7—figure supplement 1*), tumors with high Ki67 index had more elevated steady-state pH$_i$

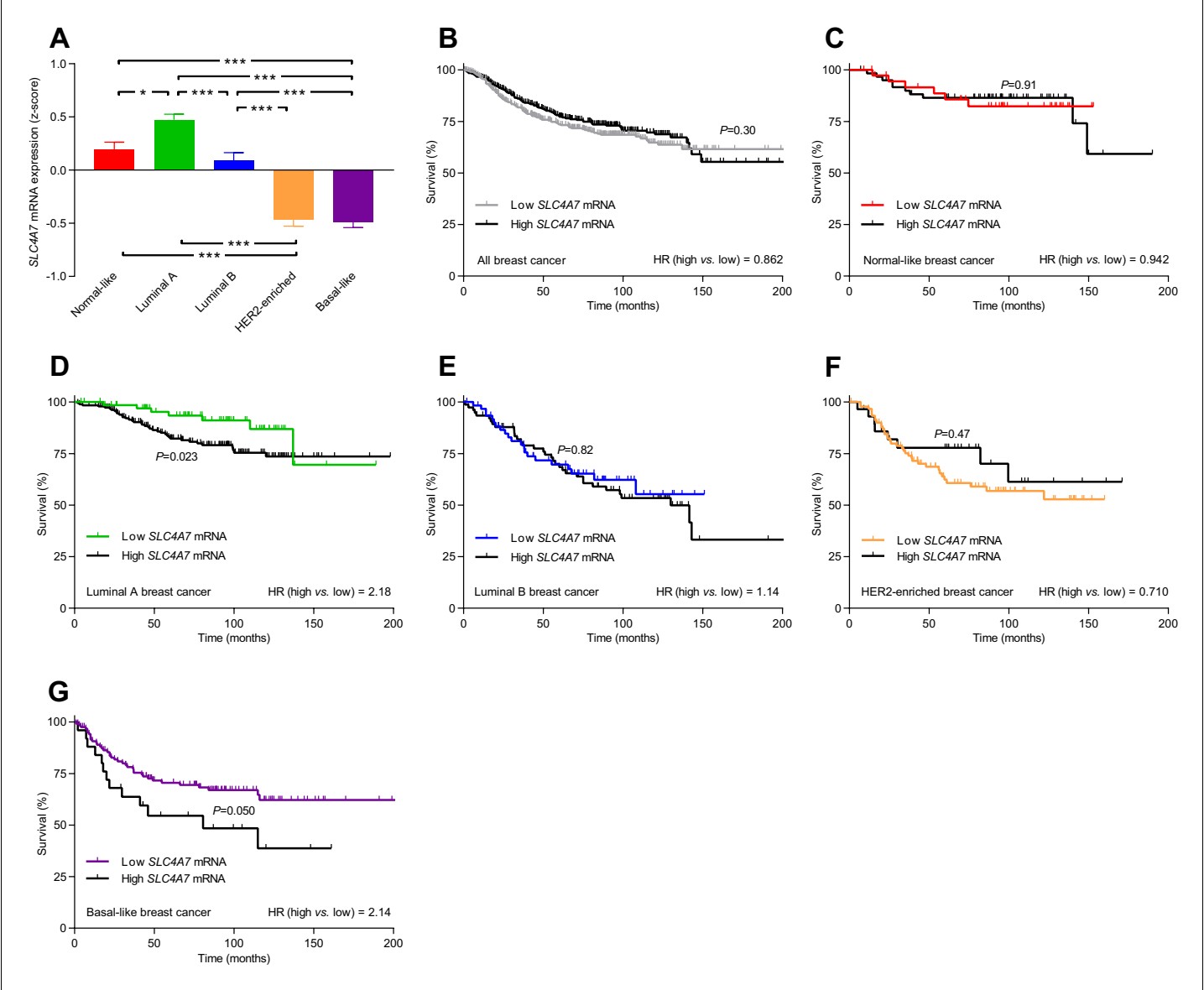

**Figure 10.** The levels of *SLC4A7* mRNA, encoding NBCn1, vary among breast cancer subtypes, and high SLC4A7 expression is associated with poor survival in patients with luminal A or basal-like breast cancer. (**A**) Variation in *SLC4A7* mRNA levels among patients with different breast cancer subtypes (n=135–344). Expression data were compared by one-way ANOVA followed by Tukey's post-test. *p<0.05, ***p<0.001. *Figure 10—source data 1* provides results from correlation analyses between *SLC4A7* mRNA expression and the mRNA levels for *ESR1*, *PGR*, *ERBB2*, *SLC9A1*, *SLC16A1*, and *SLC16A3*. (**B–G**) Survival curves stratified by *SLC4A7* mRNA levels in patients with different breast cancer molecular subtypes. The ticks on the curves represent censored subjects. Survival data were compared by Mantel-Cox and Gehan-Breslow-Wilcoxon tests. HR, hazard ratio. *Figure 10—figure supplement 1* provides results on the expression and survival consequences of *SLC16A1* mRNA. *Figure 10—figure supplement 2* provides results on the expression and survival consequences of *SLC16A3* mRNA.

The online version of this article includes the following source data and figure supplement(s) for figure 10:

**Source data 1.** The mRNA expression for *SLC4A7* correlates with that of *ESR1*, *PGR*, and *ERBB2*.

**Figure supplement 1.** The levels of *SLC16A1* mRNA, encoding MCT1, vary among breast cancer subtypes but are not associated with significant changes in patient survival.

**Figure supplement 1—source data 1.** The mRNA expression for *SLC16A1* correlates with that of *ESR1*, *PGR*, and *ERBB2*.

**Figure supplement 2.** The levels of *SLC16A3* mRNA, encoding MCT4, vary among breast cancer subtypes, and high *SLC16A3* expression is associated with poor survival except in luminal A breast cancer.

**Figure supplement 2—source data 1.** The mRNA expression for *SLC16A3* correlates with that of *ESR1*, *PGR*, and *ERBB2*.

than tumors with low Ki67 index under experimental conditions where $Na^+,HCO_3^-$ cotransport was active (*Figure 7D*). In contrast, we saw no difference in steady-state $pH_i$ between the groups in nominal absence of $CO_2/HCO_3^-$ (*Figure 7D*). The measured $CO_2/HCO_3^-$-dependent rise in $pH_i$ carried independent predictive value (odds ratio of 3.12 for a 0.1 increase in pH) to the identification of patients with elevated Ki67 index (*Figure 7G*). The rate of $pH_i$ recovery and capacity for net acid extrusion during intracellular acidification did not differ between tumors with high and low Ki67 index (*Figure 7C* and *Figure 7—figure supplement 1*).

We demonstrated elevated NBCn1 protein expression in the patient group with high compared to low Ki67 index (*Figure 7F*) although this effect was not quite significant after we adjusted for other clinicopathological characteristics (*Figure 7G*). We found no difference in NHE1 protein expression between tumors with high and low Ki67 index (*Figure 7E*).

## Increased $Na^+,HCO_3^-$ cotransport capacity and NBCn1 expression predict lymph node metastasis

Breast cancer prognosis critically depends on the invasive potential of the cancer cells, and acid-base transporters are implicated in key metastatic steps including cell migration and extracellular matrix degradation (*Boedtkjer and Pedersen, 2020*). Thus, we evaluated whether expression and function of acid-base transporters varied between primary breast cancer tissue from patients with and without lymph node metastases (*Figure 8* and *Figure 8—figure supplement 1*). We compared primary breast cancer tissue from patients without detectable tumor cells in the axillary lymph nodes to primary breast cancer tissue from patients with axillary lymph nodes containing macro-metastases, micro-metastases, or isolated tumor cells.

The initial $CO_2/HCO_3^-$-dependent $pH_i$ recovery from $NH_4^+$-prepulse-induced intracellular acidification (*Figure 8A,B*) and the corresponding $Na^+,HCO_3^-$ cotransport activity (*Figure 8C* and *Figure 8—figure supplement 1*) were accelerated in primary breast cancer tissue from patients with axillary lymph node metastases. The $Na^+,HCO_3^-$ cotransport activity quantified at $pH_i$ 6.5–6.6 carried independent predictive value to identify patients with axillary lymph node metastases (odds ratio of 1.54 for each 10 mM/min increase; *Figure 8G*). The capacity of the human primary breast carcinomas for $Na^+/H^+$ exchange activity was not significantly different between patients with and without detected lymph node metastases (*Figure 8C,G* and *Figure 8—figure supplement 1*). We also did not observe any differences in steady-state $pH_i$—in the presence or absence of $CO_2/HCO_3^-$—between primary breast carcinomas from patients with or without identified axillary lymph node metastases (*Figure 8D,G*).

In congruence with the enhanced $Na^+,HCO_3^-$ cotransport activity, we identified increased NBCn1 protein expression in primary breast cancer tissue from patients with lymph node metastases (*Figure 8F*). Thus, elevated NBCn1 protein expression was an independent predictor of lymph node metastasis with an odds ratio of 1.72 for a single-unit step increase on the applied 6-point expression scale (*Figure 8G*). The selective importance of $Na^+,HCO_3^-$ cotransport and NBCn1 expression was supported by our finding that NHE1 protein expression was reduced in primary breast cancer tissue from patients with lymph node metastases (*Figure 8E*) and qualified as an independent predictor negatively related to metastatic risk with an odds ratio of 0.60 (*Figure 8G*).

## mRNA expression levels for acid-base transporters predict patient survival

We next evaluated whether the biological implications of acid-base transporters in breast cancer tissue have consequences for patient prognosis. We used transcriptomics data to stratify a patient cohort of nearly 1500 breast cancer patients by their mRNA expression levels for acid-base transporters. We then studied how the survival proportions developed over time in the entire patient population and in subpopulations with well-defined molecular subtypes (i.e., normal-like, luminal A and B, HER2-enriched, and basal-like breast cancer). We expect no straightforward proportionality between mRNA, protein, and function when comparing across molecular subtypes or clinicopathological characteristics driven by different carcinogenic mechanisms. This should be kept in mind particularly when interpreting survival analyses performed across the whole unstratified patient population (*Figures 9B* and *10B*, and *Figure 10—figure supplements 1B* and *2B*). Focusing on individual molecular subtypes, the relationship from mRNA to protein and function is likely much

simpler, and our emphasis is therefore on the survival analyses performed after molecular subtype stratification (*Figure 9C–G*, *Figure 10C–G*, and *Figure 10—figure supplements 1C–G and 2C–G*).

The breast cancer molecular subtypes reflect differences in sex hormone (estrogen and progesterone; *Figure 9—figure supplement 1A,B*) and growth factor (HER2; *Figure 9—figure supplement 1C*) receptor expression. The individual breast cancer molecular subtypes also showed systematic differences in expression of proliferative markers (*MKI67* mRNA; *Figure 9—figure supplement 1D*) and lactate dehydrogenase involved in fermentative glycolysis (*LDHA* mRNA; *Figure 9—figure supplement 1E*), consistent with the notion that malignancy gradually increases from normal-like and luminal A across luminal B and HER2-enriched to basal-like breast cancer (*Dai et al., 2015*). In contrast, the angiogenic markers *PECAM1* and *CD34* showed the highest mRNA expression in normal-like breast cancer (*Figure 9—figure supplement 1F,G*).

The level of *SLC9A1* mRNA, encoding NHE1, was most abundant in normal-like, luminal A, and HER2-enriched breast cancer, lower in luminal B breast cancer, and lowest in basal-like breast cancer (*Figure 9A*). The *SLC9A1* mRNA expression level was not associated with altered survival for the whole population of breast cancer patients (*Figure 9B*) or for breast cancer patients suffering from normal-like (*Figure 9C*), luminal B (*Figure 9E*), HER2-enriched (*Figure 9F*), or basal-like (*Figure 9G*) breast cancer. However, patients with high *SLC9A1* mRNA expression suffering from luminal A breast cancer showed significantly improved survival (hazard ratio 0.552) compared to patients with low *SLC9A1* mRNA expression (*Figure 9D*), which is consistent with our observation of reduced lymph node metastasis among patients with high NHE1 protein expression (*Figure 8G*).

*SLC4A7* mRNA, encoding NBCn1, was expressed at the highest level in luminal A breast cancer, at intermediate level in normal-like and luminal B breast cancer, and at lowest level in HER2-enriched and basal-like breast cancer (*Figure 10A*). The high overall *SLC4A7* expression level in luminal A breast cancer argues for a prominent role of NBCn1 in this breast cancer molecular subtype and suggests that the upregulation of NBCn1 in this molecular subtype largely occurs due to transcriptional regulation. Indeed, we observed significantly shortened survival times (hazard ratio 2.18) for the luminal A breast cancer patients with the highest compared to the lowest *SLC4A7* mRNA levels (*Figure 10D*). Even though expression of *SLC4A7* mRNA was relatively low in basal-like breast cancer, we also observed a significantly worse prognosis (hazard ratio 2.14) for patients with the highest *SLC4A7* mRNA levels within this breast cancer subtype (*Figure 10G*). The importance of NBCn1 in basal-like breast cancer is likely explained by the high proliferative and glycolytic activity (*Figure 9—figure supplement 1D,E*), which will expectedly elevate the cellular acid load and thus the requirement for net acid extrusion. Shorter survival of breast cancer patients with high expression of *SLC4A7* mRNA (*Figure 10D,G*) is consistent with the greater tendency for lymph node metastasis among patients with high NBCn1 protein expression and accelerated $Na^+,HCO_3^-$ cotransport (*Figure 8G*) as well as with the enhanced proliferative activity in patients with elevated steady-state $pH_i$ due to $Na^+,HCO_3^-$ cotransport (*Figure 7G*). For patients with normal-like (*Figure 10C*), luminal B (*Figure 10E*), and HER2-enriched (*Figure 10F*) breast cancer, we saw no association of *SLC4A7* mRNA expression with survival. We also observed no overall survival differences according to the *SLC4A7* mRNA expression level across the whole unstratified population of breast cancer patients (*Figure 10B*) where the survival effects of differences in *SLC4A7* transcript levels within specific molecular subgroups are diluted in the larger patient population.

Monocarboxylate transporters play roles in eliminating lactate and associated $H^+$ from cells relying on fermentative glycolysis and allow neighboring cells to take up and metabolize the lactate through oxidation (*Pérez-Escuredo et al., 2016*). Thus, the monocarboxylate transporters could influence intra- and extracellular acid-base conditions as well as cellular energy levels in solid cancer tissue (*Boedtkjer and Pedersen, 2020*). *SLC16A1* mRNA (*Figure 10—figure supplement 1A*), encoding MCT1, and *SLC16A3* mRNA (*Figure 10—figure supplement 2A*), encoding MCT4, both showed high expression levels in basal-like breast cancer characterized by high proliferative (*Figure 9—figure supplement 1D*) and glycolytic (*Figure 9—figure supplement 1E*) activity. In contrast, *SLC16A3* mRNA expression was high (*Figure 10—figure supplement 2A*) and *SLC16A1* mRNA expression very low (*Figure 10—figure supplement 1A*) in luminal B breast cancer tissue. Notably, although *SLC16A1* mRNA levels showed no association with breast cancer survival (*Figure 10—figure supplement 1B–G*), high *SLC16A3* mRNA levels were significantly associated with—or showed tendency toward—poor survival in all breast cancer molecular subtypes (*Figure 10—figure supplement 2B,C and E–G*) except for luminal A (*Figure 10—figure supplement 2D*).

Even though the expression of the investigated acid-base transporters vary systematically across the molecular subtypes (*Figures 9A* and *10A*, *Figure 10—figure supplements 1A* and *2A*), the transcript levels for *SLC4A7*, *SLC9A1*, *SLC16A1*, and *SLC16A3* showed no significant pairwise correlation when controlled for the patterns of *ERBB2*, *ESR1*, and *PGR* mRNA expression (*Figure 9—source data 1*, *Figure 10—source data 1*, *Figure 10—figure supplement 1—source data 1* and *Figure 10—figure supplement 2—source data 1*). This observation supports that the acid-base transporters independently modify survival when analyzed for each of the breast cancer molecular subtypes separately.

## Discussion

We report here on the first large-scale study of $pH_i$ dynamics in human cancer samples. We demonstrate that acid-base transporters play key pathophysiological roles by counteracting intracellular acidification and setting the steady-state $pH_i$ in human breast cancer tissue (*Figure 1*). Furthermore, the level of cellular acidity and the capacity for net acid extrusion in human primary breast carcinomas can account for variation in proliferative activity (*Figure 7*) and are predictive for the occurrence of lymph node metastasis (*Figure 8*). The functional recordings of $pH_i$ dynamics carry predictive value that is complementary to and independent from information based on protein expression analyses (*Figures 7G* and *8G*).

We detect substantial inter-patient heterogeneity in acid-base conditions of breast carcinomas and explore the underlying modifiers and biological consequences. As illustrated in *Figure 11*, we show that (a) $Na^+,HCO_3^-$ cotransport raises $pH_i$ more in invasive lobular than ductal breast carcinomas and particularly in breast cancer tissue of high malignancy grade (*Figures 2*, *3* and *6*); (b) $Na^+/H^+$ exchange raises $pH_i$ more in estrogen receptor-negative breast carcinomas (*Figures 4* and *6*); (c) protein expression of NBCn1 and NHE1 is elevated by HER2 overexpression or gene amplification (*Figures 5* and *6*); (d) elevated steady-state $pH_i$ particularly due to $Na^+,HCO_3^-$ cotransport predicts high proliferative activity in primary breast carcinomas (*Figure 7*); (e) elevated capacity for $Na^+,HCO_3^-$ cotransport, high NBCn1 protein expression, and low NHE1 protein expression predict lymph node metastasis (*Figure 8*); and (f) high *SLC4A7* and/or low *SLC9A1* mRNA expression are associated with shorter survival in patients with luminal A and basal-like/triple-negative breast cancer (*Figures 9* and *10*).

Supporting the validity of our study, the findings are based on two distinct patient populations evaluated by separate experimental approaches: (a) the links between $pH_i$ dynamics, protein expression of acid-base transporters, and clinical and pathological patient characteristics (*Figures 1–8*) come from a cohort of 110 breast cancer patients with information derived from medical records, $pH_i$ recordings, and immunohistochemical staining; and (b) the prognostic evidence linking transcriptomics data to patient survival (*Figures 9* and *10*) comes from a separate cohort of 1457 breast cancer patients.

As depicted schematically in *Figure 1B*, $HCO_3^-$ uptake via NBCn1 is functionally equivalent to $H^+$ extrusion via NHE1 when the $CO_2/HCO_3^-$ buffer is in equilibrium (*Boedtkjer et al., 2012*). However, as summarized in *Figure 11*, it is clear from our analyses that NBCn1 and NHE1 have very different consequences in human breast cancer tissue: $Na^+,HCO_3^-$ cotransport sets the steady-state $pH_i$ associated with proliferative activity (*Figure 7G*); and the capacity for $Na^+,HCO_3^-$ cotransport and protein expression of NBCn1 predict lymph node metastasis (*Figure 8G*). In contrast, steady-state $pH_i$ in absence of $CO_2/HCO_3^-$ and the capacity for $Na^+/H^+$ exchange show no relation to proliferation (*Figure 7G*) or metastasis (*Figure 8G*); and the protein expression of NHE1 is even negatively related to axillary lymph node metastasis (*Figure 8G*). Accordingly, high *SLC4A7* but low *SLC9A1* mRNA expression is associated with poor survival in select breast cancer molecular subtypes (*Figures 9* and *10*). Although the reason for these marked differences between NBCn1 and NHE1 is not yet entirely clear, distinct cellular and subcellular expression patterns (*Boedtkjer et al., 2013*; *Lauritzen et al., 2012*), allosteric regulation by $pH_i$ and $pH_o$ (*Bonde and Boedtkjer, 2017*; *Boedtkjer et al., 2013*; *Hulikova et al., 2011*), molecular interacting partners, and responses to auto-, para-, and endocrine signals (*Boedtkjer et al., 2012*; *Boedtkjer and Aalkjaer, 2009*; *Danielsen et al., 2013*) likely play important roles.

The impact of global steady-state $pH_i$ on cell proliferation (*Figure 7G*) is consistent with earlier observations from cultured cell lines that cell cycle progression requires a permissive $pH_i$ in the

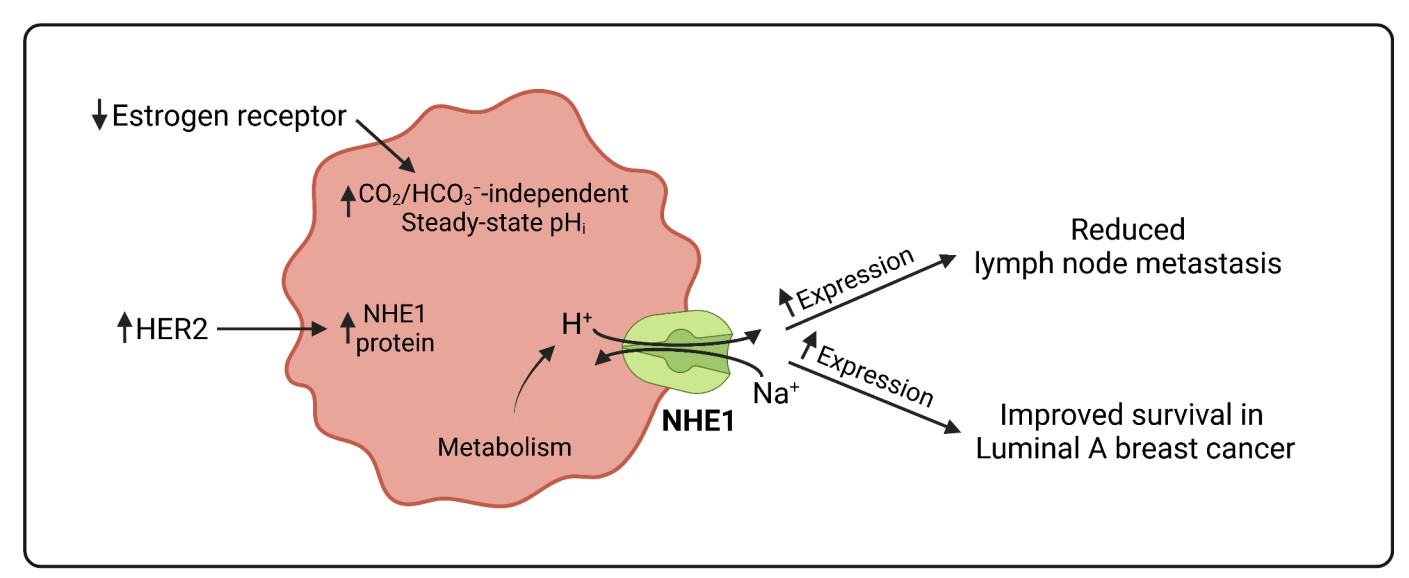

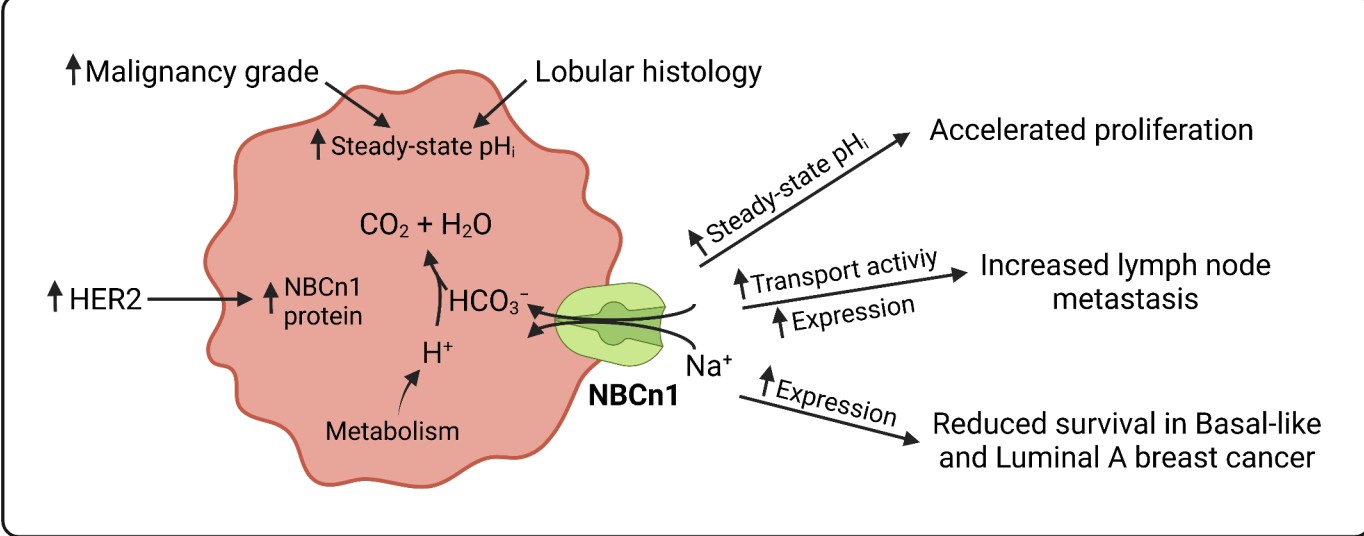

**Figure 11.** Schematics summarizing identified interactions and associated functional implications of NHE1 (upper panel) and NBCn1 (lower panel) in human breast cancer. The image was generated with BioRender.com.

slightly alkaline range (*Flinck et al., 2018*). Low $pH_i$ inhibits the enhanced DNA and protein synthesis preceding cell division although the exact molecular mechanisms have not yet been conclusively identified (*Boedtkjer, 2019*; *Boedtkjer and Aalkjaer, 2013*; *Pouysségur et al., 1985*; *Ober and Pardee, 1987*). We previously found that knockout of NBCn1 inhibits cell proliferation in murine breast cancer tissue, particularly in deep tumor regions and larger-sized tumors where fermentative glycolysis dominates and the cellular metabolic acid load is elevated (*Lee et al., 2016*; *Lee et al., 2018*).

Variation in the rate of acidic metabolic waste production and in the capacity for net acid extrusion shapes the local chemical environment and thereby influences the cancer cell phenotype important for disease progression. There is evidence that local $pH_i$ induces cytoskeletal rearrangements (*Bernstein et al., 2000*; *Pope et al., 2004*) and that cell surface pH modifies cell-cell and cell-matrix interactions (*Riemann et al., 2019*; *Stock et al., 2007*), which in combination with acid-induced degradation of the extracellular matrix (*Rofstad et al., 2006*) can promote directional migration and metastasis. Differences in cell surface pH along the axis of migration have been reported for various

migrating cells in culture based on differences in expression and activity of acid-base transporters between leading and rear ends (*Schwab et al., 2012*). We have previously found that migrating vascular smooth muscle cells generate NBCn1-dependent spatial pH gradients that are critical for directional migration (*Boedtkjer et al., 2016*), and the role of NBCn1 was subsequently supported by a study on a lung adenocarcinoma cell line (*Hwang et al., 2020*). Our current findings (*Figure 8G*) suggest that NBCn1-mediated $Na^+,HCO_3^-$ cotransport plays a similar role for the invasiveness of human breast cancer cells. Interestingly, the cellular capacity for net acid extrusion was not associated with other of the tested clinicopathological characteristics (*Figure 6C,D*), which supports that NBCn1 independently impact the metastatic potential of breast cancer cells (*Figure 8G*).

Based on the negative relationship between NHE1 expression and lymph node metastasis (*Figure 8G*) and considering the improved survival of luminal A breast cancer patients with high *SLC9A1* expression (*Figure 9D*), we propose that NHE1 is a metastasis suppressor in human breast cancer. Metastasis suppressor proteins are frequently upregulated in early cancer disease, as their expression is lost during cancer progression rather than malignant transformation (*Smith and Theodorescu, 2009*; *Guo et al., 1996*). The spatiotemporal regulation of NHE1 expression during breast cancer development has not been investigated in detail. In studies of murine carcinogen-induced breast cancer, NHE1 protein levels were unchanged in the primary cancer tissue compared to normal breast tissue and showed a tendency to decline with increasing tumor size (*Lee et al., 2016*). In previous studies on human breast cancer tissue, NHE1 protein expression was very high in ductal carcinoma in situ lesions and elevated in early primary breast carcinomas, yet showed a tendency to decrease during metastatic progression (*Boedtkjer et al., 2013*; *Lee et al., 2015*). In the same human biopsy material, NBCn1 showed a continuous trend for progressively increasing protein levels from normal breast tissue, over primary breast carcinomas, to metastatic lesions (*Boedtkjer et al., 2013*). Previous investigations in MCF7 human breast cancer cells support the functional consequences of NHE1 observed in our human breast cancer cohort: NHE1 protein expression in the MCF7 cells markedly increased during heterologous overexpression of an amino-truncated ErbB2 receptor (*Lauritzen et al., 2010*), yet pharmacological inhibition of NHE1 under these conditions stimulated cell migration (*Lauritzen et al., 2012*).

Acid-base transporters play a key role for eliminating metabolic acidic waste, and hence their pathophysiological impact depends on the metabolic activity in the specific cancer tissue. This is particularly evident in triple-negative breast cancer that shows an overall low expression of *SLC4A7* mRNA (*Figure 10A*). Nonetheless, the survival of triple-negative breast cancer patients is sensitive to variation in *SLC4A7* transcript levels (*Figure 10G*) most likely due to high proliferative and metabolic activities (*Figure 9—figure supplement 1*) that challenge $pH_i$ homeostasis. This observation supports that selectivity of anti-cancer therapies targeting acid-base transporters can be conferred not only by dramatic overexpression in cancer tissue relative to other tissues in the body but also by a greater functional dependency on net acid extrusion capacity in the cancer tissue. Pharmacological inhibitors of acid-base transporters have not yet reached clinical use. Previously described small molecule inhibitors of NBCn1 (e.g., S0859) do not provide the selectivity or pharmacokinetic properties necessary for systemic therapy (*Boedtkjer et al., 2012*; *Larsen et al., 2012*; *Steinkamp et al., 2015*); and whereas NHE1 inhibitors (e.g., cariporide, eniporide) reached phase 3 clinical studies for ischemic heart disease (*Mentzer et al., 2008*; *Zeymer et al., 2001*), they have not been sufficiently explored for cancer therapy. Recently developed inhibitors of monocarboxylate transporters show initial experimental promise as anti-cancer drugs (*Beloueche-Babari et al., 2017*).

Most current diagnostic procedures and selection of patients for targeted therapy (e.g., based on HER2 or estrogen receptors) rely on analysis of protein expression in fixed tissue. The current study validates that quantitative protein activity measurements and evaluation of functional contribution from specific molecular targets in viable tissue preparations provide additional clinically and prognostically important information. The independent information carried by RNA expression, protein expression, and acid-base transport activity suggests that considerable regulation occurs at translational and post-translational levels. The $pH_i$ dynamics in the tumor tissue can therefore not be directly deduced from data on protein or gene expression. It is especially noticeable that the protein expression levels of NBCn1 and NHE1 (*Figures 5E,F* and *6E,F*) are elevated in breast cancer tissue with HER2 overexpression or gene amplification despite low *SLC4A7* and intermediate *SLC9A1* mRNA levels (*Figures 9A* and *10A*). For NBCn1, this inverse relationship between mRNA and protein expression is in congruence with previous reports from murine breast cancer tissue where ErbB2

overexpression is associated with significantly elevated NBCn1 protein levels despite a drastically decreased *Slc4a7* mRNA level compared to normal breast tissue (*Lee et al., 2018*). Thus, increased translational activity or protein stability must be responsible for the raised NBCn1 protein levels in HER2-enriched breast cancer tissue. Because the *SLC4A7* mRNA level—based on the abovementioned considerations—does not reflect the NBCn1 protein level or functional capacity in HER2-enriched breast cancer, the survival analysis based on *SLC4A7* mRNA expression (*Figure 10F*) should be interpreted with caution for this molecular subtype. Even without a link between *SLC4A7* mRNA expression and survival, the NBCn1 protein expression level and functional capacity could influence patient prognosis.

In light of the widespread use of HER2-targeted breast cancer therapy, it is intriguing that HER2 signaling shows apparently opposing effects on breast cancer progression by concurrently upregulating the protein expression of NBCn1 and NHE1 (*Figures 5E,F* and *8G*). Our findings indicate that (a) additional detailed analysis of downstream signaling effects (e.g., relative upregulation of NBCn1 vs. NHE1) may provide more accurate predictive value than simple evaluation of HER2 overexpression and gene amplification and (b) selective targeting of individual HER2-activated effectors—including acid-base transporters—could optimize the current therapeutic approach based on direct HER2 inhibition.

Luminal A breast cancer differs from other breast cancer molecular subtypes in the profile whereby expression of acid-base transporters influences mortality rates. Elevated mRNA expression of *SLC4A7* impedes (*Figure 10D*), *SLC9A1* improves (*Figure 9D*), whereas *SLC16A1* and *SLC16A3* show no influence on (*Figure 10—figure supplements 1D* and *2D*) patient survival in luminal A breast cancer. The observation regarding *SLC16A3* is notable because high expression worsens (or tends to worsen) patient prognosis in the other breast cancer molecular subtypes (*Figure 10—figure supplement 2C and E–G*). Luminal A breast cancer comprises 50–70% of breast cancer cases in the United States and Europe (*Kulkarni et al., 2019*; *Acheampong et al., 2020*; *Valla et al., 2016*) and is clinically interesting because distant metastasis occurs throughout follow-up for as long as 25 years after initial diagnosis (*Yu et al., 2019*). This is different from, for instance, luminal B breast cancer where the risk of metastasis is high the first 5 years after diagnosis but then markedly declines (*Yu et al., 2019*). Given the identified link between acid-base conditions and metastasis (*Figure 8*), the distinct prognostic dependency of luminal A breast cancer patients on expression of acid-base transporters could hold a key to the underlying pathophysiology resulting in protracted metastatic risk.

Invasive lobular carcinomas show diffusely infiltrative growth patterns that differ from the dominant masses typical for invasive ductal carcinomas (*Thomas et al., 2019*). Lost expression of the cell-cell adhesion molecule E-cadherin can explain the tendency for cancer cells from lobular carcinomas to invade in single file (*Gamallo et al., 1993*; *Pai et al., 2013*), which is more difficult to delineate by mammography, and thus account for the higher positive surgical margin rates following lumpectomy procedures (*van Deurzen, 2016*). More efficient net acid extrusion in lobular compared to ductal breast carcinomas is evident from the greater elevation of $pH_i$ by $Na^+,HCO_3^-$ cotransport (*Figures 2* and *6*) and likely translates to enhanced extracellular acidification. Thus, based on the downregulation of E-cadherin expression in response to extracellular acidosis—previously identified in cultured cell lines (*Riemann et al., 2019*; *Suzuki et al., 2014*)—it is an intriguing possibility that cellular acid-base handling in human breast carcinomas shapes the histology characteristic growth patterns.

To our knowledge, the current study reports from the largest existing human cancer cohort (*Table 1*) with detailed cellular acid-base information. Still, the size of the cohort comes with some limitations. Mortality rates for breast cancer patients undergoing breast-conserving surgery are generally low (*Onitilo et al., 2015*). Therefore, at currently 2–5 years of follow-up, we are unable to perform meaningful survival analyses linking directly to measurements of pH or protein expression. However, our meta-analysis based on transcriptomics data from larger cohorts with longer follow-up partly compensates for this limitation (*Figures 9* and *10*, and *Figure 10—figure supplements 1* and *2*). In addition, despite the large overall cohort size, we are for the individual subgroup analyses limited by the natural incidence of specific clinicopathological characteristics. For some of the less frequent characteristics—e.g., rarer histological types (*Table 1*)—the subgroups are too small for statistical comparison.

Whereas the short delay from tissue isolation to functional evaluation is a unique strength of the current study—because it minimizes the risk of phenotypical changes and thereby strengthens the

connection to the clinical condition—it limits the experimental possibilities for detailed mechanistic and molecular studies. The experimental setup allows for manipulation and precise control of the buffer compositions; but with half-lives of protein degradation of 76 and 48 hr for NBCn1 and NHE1, respectively (**Olesen et al., 2018**), we cannot realistically reduce overall cellular protein levels by interfering with their expression—e.g., by RNAi knockdown technologies (**Boedtkjer et al., 2006**)—in the human biopsy material. Also, the available pharmacological options are too unspecific and without selectivity for individual $Na^+,HCO_3^-$ cotransporters (**Boedtkjer et al., 2016**; **Boedtkjer et al., 2012**; **Larsen et al., 2012**). Still, our previous studies confirm that the $Na^+$, $HCO_3^-$ cotransport in human breast cancer tissue is of low 4,4′-diisothiocyano-2,2′-stilbenedisulfonic acid sensitivity (**Boedtkjer et al., 2013**), which is a pharmacological characteristic of NBCn1 relative to other $Na^+,HCO_3^-$ cotransporters (**Boedtkjer et al., 2012**; **Choi et al., 2000**; **Romero et al., 2004**). This pharmacological profile thus corroborates strong recent molecular evidence—based on gene knockout technology—that the upregulated $Na^+,HCO_3^-$ cotransport in two different murine breast cancer models completely depends on NBCn1 (**Lee et al., 2016**; **Lee et al., 2018**).

In conclusion, we identify distinct patterns of $pH_i$ dynamics as well as mRNA and protein expression of acid-base transporters among breast cancer patients based on clinical and pathological characteristics and molecular subtypes. The mechanisms of acidic waste product elimination reflect the heterogeneity in human breast cancer tissue. Dependency on $Na^+,HCO_3^-$ cotransport for steady-state $pH_i$ regulation independently predicts proliferative activity, whereas the capacity for $Na^+$, $HCO_3^-$ cotransport activity and the expression of NBCn1 predict lymph node metastasis and patient survival. In contrast, NHE1 expression negatively predicts lymph node metastasis and patient survival. Together, these findings underscore the important pathophysiological role of acid-base homeostasis in human breast cancer tissue and emphasize the potential of acid-base transporters as anti-cancer targets.

# Materials and methods

**Key resources table**

| Reagent type (species) or resource | Designation | Source or reference | Identifiers | Additional information |
|---|---|---|---|---|
| Gene (*Homo sapiens*) | *SLC4A7* | GenBank | Gene ID: 9497 | Encodes NBCn1 |
| Gene (*Homo sapiens*) | *SLC9A1* | GenBank | Gene ID: 6548 | Encodes NHE1 |
| Gene (*Homo sapiens*) | *SLC16A1* | GenBank | Gene ID: 6566 | Encodes MCT1 |
| Gene (*Homo sapiens*) | *SLC16A3* | GenBank | Gene ID: 9123 | Encodes MCT4 |
| Biological sample (*Homo sapiens*) | Surgical breast biopsies | Regionshospitalet Randers, Denmark | This study cohort | Cancer and matched normal tissue |
| Chemical compound, drug | Collagenase type 3 | Worthington Biochemical Corporation | Cat. #: LS004182 | 450 IU/mL |
| Chemical compound, drug | BCECF-AM | Thermo Fisher Scientific | Cat. #: B1170 | 3 µM |
| Antibody | Anti-NBCn1 (Rabbit polyclonal) | Jeppe Praetorius, Aarhus University, Denmark **Damkier et al., 2006** | Reference | IHC (1:100) |
| Antibody | Anti-NHE1 (Mouse monoclonal) | Santa Cruz Biotechnology | Cat. #: sc-136239; RRID:AB_2191254 | IHC (1:100) |
| Commercial assay, kit | OptiView DAB IHC detection kit | Roche Diagnostics | RRID:AB_2833075 | Goat anti-rabbit and anti-mouse |
| Software, algorithm | SPSS | IBM | RRID:SCR_002865 | |
| Software, algorithm | Prism | GraphPad | RRID:SCR_002798 | Version 9.1.1 |

## Human breast biopsies

Viable tissue biopsies of human breast cancer tissue and matched normal breast tissue were obtained from breast-conserving lumpectomies at the Department of Surgery, Regionshospitalet Randers, Denmark, essentially as previously described (*Boedtkjer et al., 2013*; *Lee et al., 2015*). We collected samples only from primary breast cancer resections, that is, no samples were acquired from recurrent tumors or metastatic sites. None of the included patients had received pre-operative radiation or chemotherapy. The studies included women, who were at least 18 years of age and presented with operable primary breast cancer (>10 mm) diagnosed by triple test including clinical examination, mammography combined with ultrasonography, and fine-needle aspiration cytology and/or core-needle biopsy. *Table 1* summarizes the clinical and pathological characteristics of the 110 included patients. We obtained information regarding patient age, tumor size, histology, malignancy grade, expression of estrogen receptors, HER2 overexpression or gene amplification, Ki67 index, and lymph node metastasis from the medical records of the standard diagnostic care.

## Preparation of freshly isolated organoids

We freshly isolated epithelial organoids—multicellular conglomerates of approximately 150 μm diameter dominated by cytokeratin-19-positive epithelial cells (*Lee et al., 2016*; *Lee et al., 2018*; *Lee et al., 2015*)—from breast cancer tissue and normal breast tissue through partial collagenase digestion of the collected breast biopsies (*Figure 1A,B*). The tissue samples were first cut into 1 mm pieces in phosphate-buffered saline and then transferred to Tissue Culture Flat Tubes (Techno Plastic Products AG, Switzerland) containing advanced DMEM/F12 culture medium (Life Technologies, Denmark) added 10% fetal bovine serum (Biochrom AG, Germany), 1% GlutaMAX (Thermo Fisher Scientific, Denmark) and 450 IU/mL collagenase type 3 (Worthington Biochemical Corporation, Lakewood, NJ). After continuous overnight shaking at 60 rpm in an incubator with 5% $CO_2$ at 37°C, the isolated organoids sedimented for 20 min by gravitational forces and were then used directly for experiments without culture in order to best retain the functional characteristics of the sampled breast tissue.

## pH$_i$ measurements

We studied pH$_i$ dynamics in epithelial organoids—freshly isolated from human breast cancer tissue and normal breast tissue—loaded with the pH-sensitive fluorophore 2',7'-bis-(2-carboxyethyl)-5-(and-6)-carboxyfluorescein (BCECF). We added 3 μM acetoxymethyl ester form of BCECF to a physiological saline solution containing 0.1% dimethyl sulfoxide, and loaded the organoids at 37°C for approximately 20 min before they were investigated on the stage of a Nikon Diaphot 200 microscope (Nikon, Japan) equipped with an SRV CCD Retiga camera (QImaging, Canada) and VisiView software (Visitron Systems, Germany). Emission light was collected at 530 nm during alternating excitation at 440 and 495 nm. After background subtraction, the $F_{495}/F_{440}$ BCECF fluorescence ratio was calibrated to pH$_i$ using the high-[K$^+$] nigericin technique (*Thomas et al., 1979*).

Intracellular acidification was induced with the $NH_4^+$-prepulse technique (*Boron and De Weer, 1976*). Using the equation: $\beta = \Delta[NH_4^+]/\Delta pH_i$, we calculated the intracellular intrinsic buffering capacity from the change in pH$_i$ upon addition or washout of $NH_4Cl$ under $CO_2/HCO_3^-$-free conditions. The contribution of $CO_2/HCO_3^-$ to the intracellular buffering power was calculated based on the formula $\beta CO_2/HCO_3^- = 2.3 \cdot [HCO_3^-]_i$ (*Roos and Boron, 1981*). Concentrations of $NH_4^+$ and $HCO_3^-$ were computed from the Henderson-Hasselbalch equation. We calculated net acid extrusion activity as the product of the total intracellular buffering capacity and the rate of pH$_i$ recovery from $NH_4^+$-prepulse-induced intracellular acidification. Net acid extrusion activities were generally calculated for 30 s intervals covering the full pH$_i$ recovery phase; but in cases with extreme intracellular acidification (reaching below pH$_i$ 6.4), we omitted the most acidic phase while maintaining a pH$_i$ span of no less than 0.1. We plotted net acid extrusion as function of the average pH$_i$ within the individually analyzed intervals and calculated for each experiment the transport rate at pH$_i$ 6.5–6.6. A few organoids were excluded from the analyses because of insufficient $NH_4^+$-prepulse-induced acidification.

The $CO_2/HCO_3^-$-containing physiological saline solution contained (in mM): 119 NaCl, 22 NaHCO$_3$, 10 HEPES, 1.2 MgSO$_4$, 2.82 KCl, 5.5 glucose, 1.18 KH$_2$PO$_4$, 0.03 EDTA, 1.6 CaCl$_2$. $CO_2/HCO_3^-$-free solutions were produced by substitution of $HCO_3^-$ with Cl$^-$, whereas Na$^+$-free solutions were produced by substitution of Na$^+$ with *N*-methyl-D-glucammonium (NMDG$^+$), except for the

$NaHCO_3$ that was substituted with choline-$HCO_3$. All solutions contained 5 mM probenecid to inhibit the organic anion transporter and avoid extrusion of BCECF from the cancer cells (*Lee et al., 2015*). The buffer solutions were aerated with 5% $CO_2$/balance air (for $CO_2$/$HCO_3^-$-containing solutions) or nominally $CO_2$-free air (for $CO_2$/$HCO_3^-$-free solutions) and pH adjusted to 7.40 at 37˚C.

## Immunohistochemistry

Histological sections were prepared from paraffin-embedded tissue blocks and immunohistochemically stained on a BenchMark ULTRA automated staining system (Roche Diagnostics, Indianapolis, IN). After deparaffinization, slides were heated to 100˚C and pretreated using BenchMark ULTRA CC1 conditioning solution (Roche Diagnostics). The slides were then incubated with primary antibody diluted 1:100 in Dako REAL Antibody Diluent (S2022; Agilent Technologies, Inc, Santa Clara, CA) for 32 min. The rabbit anti-$NH_2$-terminal NBCn1 antibody was generously provided by Dr Jeppe Praetorius (*Damkier et al., 2006*). The mouse anti-NHE1 antibody (#sc-136239, RRID: AB_2191254) was from Santa Cruz Biotechnology (Dallas, TX). Finally, endogenous peroxidase activity was inhibited, and bound antibody was detected with the OptiView DAB IHC detection kit (Roche Diagnostics, RRID:AB_2833075). Stained slides were imaged with a Hamamatsu NanoZoomer S60 digital slide scanner (Japan). Staining intensity was scored by an experienced breast pathologist.

Information on estrogen receptor, HER2, and Ki67 expression was obtained from the standard diagnostic procedures. We observed a clear distinction between one group with very high estrogen receptor expression (≥90% positive cells) and another group with low estrogen receptor expression (≤15% positive cells) similar to values (median of 10%) previously reported for normal breast tissue (*Oh et al., 2017*). The absence of intermediate expression levels provided a clear and obvious separation between the groups. As previously noted by others (*Coates et al., 2015*), Ki67 expression displays a continuous distribution with no clear separation between groups of high and low expression. The median Ki67 index was 20% in our study cohort as a whole and also in the patient group with estrogen receptor-positive disease (see *Figure 1—source data 1*). Therefore, we followed the guidelines of the 2015 St Gallen International Expert Consensus recommending a cut-off setting of 30% for identification of patients with a clearly high Ki67 index (*Coates et al., 2015*).

## Transcript levels and survival data in human breast cancer

We retrieved seven normalized microarray datasets from studies by van de Vijver and co-workers (*van de Vijver et al., 2002*), Guo and co-workers (*Guo et al., 2005*), Calza and co-workers (*Calza et al., 2006*), and from the Gene Expression Omnibus series: GSE1992 (*Hu et al., 2006*), GSE2034 (*Wang et al., 2005*), GSE11121 (*Schmidt et al., 2008*), and GSE3143 (*Bild et al., 2006*). These studies collectively cover 1457 breast cancer patients with information on our genes of interest. The majority of the studies measure gene expression with multiple probes per gene; and therefore, we collapsed the multiple expression values by gene symbol using the maximum mean probe intensity. Next, we assigned each sample to one of the five well-defined breast cancer molecular subtypes (normal-like, luminal A, luminal B, HER2-enriched, and basal-like) identified in previous studies (*Sørlie et al., 2001*; *Perou et al., 2000*) using the PAM50 Breast Cancer Intrinsic Classifier (*Parker et al., 2009*).

To conduct a meta-analysis, we cross-sample standardized each dataset separately, and then combined all seven datasets into one expression matrix that was subjected to a second round of cross-sample standardization. We used this standardized expression matrix to compare and test for differential gene expression levels of *CD34*, *ESR1*, *HER2*, *LDHA*, *MKI67*, *PECAM1*, *PGR*, *SLC4A7*, *SLC9A1*, *SLC16A1*, and *SLC16A3* between the five molecular subtypes and for survival analyses. Kaplan-Meier survival curves were constructed for groups with high and low mRNA expression defined by z-scores above 0.5 and below –0.5, respectively. In cases where either the group of high or low mRNA expression included less than 10 deaths, we used z-score cut-off thresholds of 0.3 and –0.3. For z-score calculation, the difference between the raw score and the population mean was divided by the population standard deviation.

## Statistics

Data are given as mean ± SEM unless otherwise specified; n equals number of patients (i.e., biological replicates) and experiments were performed one time for each experimental condition. No

explicit power calculations were performed. Within the study periods, tissue was sampled from all patients, who fulfilled the inclusion criteria, and allocated to the appropriate groups based on information from the standard diagnostic procedures. The investigators were blinded during the experiments and analyses, as the clinicopathological information was not collected until later. We considered p-values smaller than 0.05 as statistically significant. Acid-base transport activities and steady-state $pH_i$ values were compared between groups by mixed-effects analyses followed by Sidak's multiple comparisons test. We compared categorical protein expression levels by $\chi^2$-tests for trend. mRNA expression data were compared between multiple groups by one-way ANOVA followed by Tukey's post-test. Relationships between patient age, tumor size, acid-base transport activity, and steady-state $pH_i$ were tested by Spearman's correlation analyses. We identified associations between multiple clinical and pathological independent variables and continuous dependent variables based on multiple linear regression analyses, whereas we identified associations with dichotomous and ordered categorical dependent variables using binomial and ordinal logistic regression analyses, respectively. Kaplan-Meier survival curves were compared by Mantel-Cox and Gehan-Breslow-Wilcoxon tests. We tested pairwise correlations between mRNA expression of acid-base transporters, angiogenic markers, and sex hormone and growth factor receptors based on Pearson and partial correlation analyses. Statistical analyses were performed with GraphPad Prism 9.0.0 (RRID: SCR_002798) and IBM SPSS Statistics (RRID:SCR_002865) software.

### Study approval

All participants gave written informed consent. The Mid-Jutland regional division of the Danish Committee on Health Research Ethics (M-20100288) and the Danish Data Protection Agency (1-16-02-191-16) approved the procedures for tissue sampling and data handling, respectively.

## Acknowledgements

The authors would like to thank doctors and nurses, especially Dr Ida E Holm, at Regionshospitalet Randers for their contribution to tissue collection and pathology evaluation. We thank Jette K Jensen and Karen L Kristensen, Department of Pathology, Regionshospitalet Randers, for expert technical assistance. Funding: This work was financially supported by the Independent Research Fund Denmark (7025-00050B to Boedtkjer), the Novo Nordisk Foundation (NNF18OC0053037 to Boedtkjer), and the Danish Cancer Society (R136-A8670 to Toft).

## Additional information

### Competing interests

Ebbe Boedtkjer: is an inventor on patents covering NBCn1 as target for cancer therapy (EP 3271402). The other authors declare that no competing interests exist.

### Funding

| Funder | Grant reference number | Author |
| --- | --- | --- |
| Sundhed og Sygdom, Det Frie Forskningsråd | 7025-00050B | Ebbe Boedtkjer |
| Novo Nordisk Fonden | NNF18OC0053037 | Ebbe Boedtkjer |
| Danish Cancer Society | R136-A8670 | Nicolai J Toft |

The funders had no role in study design, data collection and interpretation, or the decision to submit the work for publication.

### Author contributions

Nicolai J Toft, Data curation, Formal analysis, Investigation, Methodology, Writing - review and editing; Trine V Axelsen, Data curation, Formal analysis, Investigation, Writing - review and editing; Helene L Pedersen, Mark Burton, Mads Thomassen, Data curation, Formal analysis, Writing - review and editing; Marco Mele, Eva Balling, Peer M Christiansen, Data curation, Writing - review and editing;

Tonje Johansen, Data curation; Ebbe Boedtkjer, Conceptualization, Formal analysis, Supervision, Funding acquisition, Visualization, Methodology, Writing - original draft, Project administration, Writing - review and editing

### Author ORCIDs
Marco Mele http://orcid.org/0000-0002-8156-7804
Ebbe Boedtkjer https://orcid.org/0000-0002-5078-9279

### Ethics
Human subjects: All participants gave written informed consent. The Mid-Jutland regional division of the Danish Committee on Health Research Ethics (M-20100288) and the Danish Data Protection Agency (1-16-02-191-16) approved the procedures for tissue sampling and data handling, respectively.

### Decision letter and Author response
Decision letter https://doi.org/10.7554/eLife.68447.sa1
Author response https://doi.org/10.7554/eLife.68447.sa2

## Additional files

### Supplementary files
• Transparent reporting form

### Data availability

In order to comply with the ethical approval, we share the human data presented in Figure 1–8 and corresponding figure supplements (data on acid-base transport activity, intracellular pH, and protein expression of transporters linked to clinicopathological information) in de-identified form. Following consultation with the legal team at the Regional Committee on Health Research Ethics, we have generated dataset files where restricted information is grouped in intervals each consisting of no less than five individuals. To provide the reader with the best possible data insight, we also show figure supplements with more detailed and advanced plots of the data and include the corresponding de-identified dataset. The meta analyses presented in Figure 9, 10, and corresponding figure supplements (data on RNA expression linked to patient survival) are based on data that have previously been published by other investigators, as detailed in the manuscript and the dataset list.

The following previously published datasets were used:

| Author(s) | Year | Dataset title | Dataset URL | Database and Identifier |
| --- | --- | --- | --- | --- |
| van de Vijver MJ, He YD, van't Veer LJ, Dai H, Hart AA, Voskuil DW, Schreiber GJ, Peterse JL, Roberts C, Marton MJ, Parrish M, Atsma D, Witteveen A, Glas A, Delahaye L, van der Velde T, Bartelink H, Rodenhuis S, Rutgers ET, Friend SH, Bernards R | 2002 | A gene-expression signature as a predictor of survival in breast cancer | http://bioconductor.org/packages/release/data/experiment/html/cancer-data.html | Budczies J, cancerdata |
| Calza S, Hall P, Auer G, Björle J, Klaar S, Kronenwett U, Liu ET, Miller L, Ploner A, Smeds J, Bergh J, Pawitan Y | 2006 | Intrinsic molecular signature of breast cancer in a population-based cohort of 412 patients | https://www.meb.ki.se/sites/yudpaw/wp-content/uploads/sites/5/papers/Intrinsic_Signature.zip | Karolinska Institutet, Intrinsic_Signature.zip |

| Wang Y, Klijn JGM, Zhang Y, Sieuwerts AM, Look MP, Yang F, Talantov D, Timmermans M, van Gelder MEM, Yu J, Jatkoe T, Berns EMJJ, Atkins D, Foekens JA | 2006 | The molecular portraits of breast tumors are conserved across microarray platforms | https://www.ncbi.nlm.nih.gov/geo/query/acc.cgi?acc=GSE1992 | NCBI Gene Expression Omnibus, GSE1992 |
|---|---|---|---|---|
| Wang Y, Klijn JGM, Zhang Y, Sieuwerts AM, Look MP, Yang F, Talantov D, Timmermans M, van Gelder MEM, Yu J, Jatkoe T, Berns EMJJ, Atkins D, Foekens JA | 2005 | Gene-expression profiles to predict distant metastasis of lymph-node-negative primary breast cancer | https://www.ncbi.nlm.nih.gov/geo/query/acc.cgi?acc=GSE2034 | NCBI Gene Expression Omnibus, GSE2034 |
| Schmidt M, Böhm D, von Törne C, Steiner E, Puhl A, Pilch H, Lehr H-A, Hengstler JG, Kölbl H, Gehrmann M | 2008 | The humoral immune system has a key prognostic impact in node-negative breast cancer | https://www.ncbi.nlm.nih.gov/geo/query/acc.cgi?acc=GSE11121 | NCBI Gene Expression Omnibus, GSE11121 |
| Bild AH, Yao G, Chang JT, Wang Q, Potti A, Chasse D, Joshi MB, Harpole D, Lancaster JM, Berchuck A, Olson Jr JA, Marks JR, Dressman HK, West M, Nevins JR | 2006 | Oncogenic pathway signatures in human cancers as a guide to targeted therapies | https://www.ncbi.nlm.nih.gov/geo/query/acc.cgi?acc=GSE3143 | NCBI Gene Expression Omnibus, GSE3143 |

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
