## [Decision Letter]

**Acceptance summary:**

Dysregulated acid-base microenvironment has been revealed to be associated with several adverse outcomes, such as drug resistance and enhanced aggression. In this research a large-scale cohort comprised of patient-derived organoids were employed mimicking exactly the condition under clinical setting to observe effect of dysregulated acid-base transporter on clinical parameters and survival. This work provides potential novel therapeutic avenue through targeting acid-base transporter regulating pH.

**Decision letter after peer review:**

Thank you for submitting your article "Acid-base transporters and pH dynamics in human breast carcinomas predict proliferative activity, metastasis, and survival" for consideration by *eLife*. Your article has been reviewed by 3 peer reviewers, including Caigang Liu as the Reviewing Editor and Reviewer #1, and the evaluation has been overseen by Maureen Murphy as the Senior Editor.

Essential revisions:

1) Most of the conclusions could be well supported by clinical data statistics. However, there still exist some concerns expected to be explained, especially, detailed mechanism or therapeutic strategies description regarding pH dysregulation are deficient.

2) There are several contradictory and confusing conclusions in the manuscript remaining to be discussed. It would be better to reorganize the manuscript in a clearer and more convincing manner to present the data in the study.

*Reviewer #1:*

The authors launched an observational study on 110 breast cancer organoids, measured the cancer and normal cell net acid extrusion capacity to correlate different clinicopathological parameters with intracellular pH regulator (NHE1 and NBCn1) level, and unravel the relationship between them and tumor proliferation, lymph node metastasis together with the impact of cotransporter mRNA on patient's survival, thus highlighting the role of pH regulator in malignancy. This study was performed on a large-scale cohort based on tissue samples to comprehensively reflect the heterogeneity setting. However, there lacks detailed mechanism or therapeutic strategies description regarding pH dysregulation. The data has been sufficient to hold back the views proposed and the results will provide better understanding for the role of dysregulated pH in cancer progression and help future studies regarding targeting pH regulator in managing cancer.

1. In Figure 9 and 10, SLC9A1 and SLC4A7 mRNA level was analyzed for their correlation with survival outcome, it would be better to supplement survival analysis regarding the expression of SLC9A1 and SLC4A7 protein based on your study cohort.

2. Please specify the reasons for ER and Ki-67 cut-off value setting and include them in Method or insert supporting reference literature.

3. It is described in lines 320-321 that the protein expression level of NHE1 and NBCn1 was not significantly influenced by estrogen receptor status. And it is described in lines 333-335 that the protein expression levels of NHE1 as well as NBCn1 were elevated in breast carcinomas with HER2 overexpression or gene amplification both before and after adjustment for other clinicopathological characteristics. But the survival figures of Figure 9, 10 indicated that 'For patients with HER2-enriched breast cancer, we saw no association of SLC4A7 or SLC9A1 mRNA expression with survival; The high overall SLC4A7 expression level in Luminal A breast cancer suggests a prominent role of NBCn1 in this breast cancer molecular subtype. And patients with high SLC9A1 mRNA expression suffering from Luminal A breast cancer showed significantly improved survival (hazard ratio 0.552) compared to patients with low SLC9A1 mRNA expression'. The conclusions seem to be contradictory, please explain this.

4. In lines 75-81: 'In human and murine breast cancer tissue analyzed without stratification by molecular subtype, Na+, HCO_3_^-^ cotransport activity is elevated and protein expression of the NBCn1 (SLC4A7) and MCT1 (SLC16A1) and MCT4 (SLC16A3) are upregulated compared to normal breast tissue. Protein expression of the NHE1 (SLC9A1) is either unchanged or moderately upregulated when breast cancer tissue as one unstratified group is compared to normal breast tissue.' However, worse prognosis was observed in patients suffering from Luminal A or Basal-like/triple-negative breast cancer with high SLC4A7 and/or low SLC9A1 mRNA expression. The expression level of NHE1 (SLC9A1) and the prognostic data are conflicting, please discuss it in detail.

5. The authors found that the expressions of NBCn1 and NHE1 were related to lymph node metastasis and cancer cell proliferation. However, lymph node metastasis and cancer cell proliferation are both key factors predicting worse prognosis. The existing data cannot indicate whether intratumor PH directly affects the patient's prognosis, or indirectly affects the prognosis through lymph nodes metastasis or cancer cell proliferation. Please add survival charts of patients in different subgroups to illustrate it.

6. The authors separately verified the effects of NBCn1 (SLC4A7), MCT1 (SLC16A1), MCT4 (SLC16A3) and NHE1 (SLC9A1) on the prognosis of patients, but whether there exist synergistic or antagonistic effect among the four genes, or whether there is mutual regulation relationship should be addressed.

7. This manuscript requires a more comprehensive summary regarding previous reported role of abnormal pH in promoting cancer in Introduction, and the same about mechanism description and future interfering strategies in Discussion to help readers comprehend this field more conveniently.

8. Breast cancer organoids are mainly composed of cytokeratin 19-positive epithelial cells while the authors only provide the microscopic pictures. It would be better to provide immunofluorescence or IHC staining pictures of cytokeratin 19-positive epithelial cells.

9. Several references date 20 years ago, new references supporting the authors' findings would strengthen the hypothesis. Meanwhile, please arrange it in accordance with the reference format required by *eLife*.

*Reviewer #2:*

In this extensive study, authors have shown that Na+,HCO_3_^-^ cotransport and Na+/H^+^-exchange transporters predominantly mediate overall cellular acid extrusion in breast cancer microenvironment. In addition, authors demonstrate that higher activity of Na+,HCO_3_^-^-cotransport correlates with increased growth and lymph node metastasis of breast cancer. The mechanistic insights gained from the study will help in the development of therapeutic strategies for breast cancers. The strengths of this study include the use of organoid culture system, and detailed cellular acid-base homeostasis analyses in an extensive cohort of breast cancer patients. Overall, this study is of interest to the peers working in the breast cancer field.

1. It would be interesting if the authors have used genetic knockdown or specific inhibitors against NBCn1 and NHE1 in breast cancer organoids to avoid the redundant functions of other ion exchangers.

2. Did the authors analyze the expression of angiogenic markers such as CD31 or CD34 in the patient samples?

3. For the ease of readers, in Figure 1B, authors should also highlight the observations on expression and correlation of NBCn1 and NHE1 with overall survival in different breast cancer subtypes.

4. Authors should discuss the possible limitations of their study in the Discussion part of the manuscript.

5. Please correct the spelling of Maier to Kaplan-Meier in methods.

6. In Figure 1 E, authors should use only one asterisk to show the statistical significance between the groups.

*Reviewer #3:*

Nicolai J. Toft et al. investigated the feature of acid-base metabolism in breast cancer tissue from several aspects, revealing the underlying mechanism of pH regulation according to the heterogeneity. Moreover, the acid-base transport and intracellular pH state were proved capable of predicting the proliferation ability and lymph node metastasis risk of breast cancer, and were associated with the survival in specific breast cancer subtypes. The conclusions of this paper are mostly well supported by data. However, it would have been clearer and more convincing to present the clinical significance combining all data in this study, especially in a more direct way (using tables or graphs).

1. An exposition for combined analysis of the results in figure 1-6 and figure 7-8 is expected to let the readers better understand the clinical significance (like prediction for proliferation and lymph node metastasis) of the various situations in breast cancer with different acid-base metabolism features.

2. There are some questions of how to select different indicators to evaluate the prognosis tendency. For example, data in figure 5E-F and 6E-F showed that Her2 receptor increased the expression of both NBCn1 and NHE1, which exert opposite predictive function in lymph node metastasis according to the data in figure 8G. In this condition, how to evaluate the risk of metastasis? What is the clinical significance of the data on Her2-enriched subtype?

3. How to confirm the statistical significance of the difference in the figures showing "traces of NH^+^4-prepulse-induced pHi dynamics" (like figure 2A-B)? Meanwhile, the p value is better to be marked in the figures.

4. The standard to define estrogen receptor-negative breast cancer and estrogen receptor-positive breast cancer should be pointed out. The results in figure 4 is confusing.

5. The information in figure 10A and 10F showed that the mRNA of SLC4A7, which encodes protein NBCn1, expressed in a very low level and could not predict the survival prognosis in HER2-riched breast cancer subtype. However, in figure 5F and figure 6F, the data indicated that HER2 expression level was positively associated with higher NBCn1 frequency. So, a further explanation is needed for the contradiction of these data and their clinical significance.

6. The authors should add more biological and molecular experiments to better support the conclusions.

---

## [Author Response]

Essential revisions:1) Most of the conclusions could be well supported by clinical data statistics. However, there still exist some concerns expected to be explained, especially, detailed mechanism or therapeutic strategies description regarding pH dysregulation are deficient.

We have now very substantially expanded the introduction (l. 82-99), results (l. 380-387), and discussion (l. 452-469 and 486-565) sections in order to better explain how our findings relate to molecular mechanisms of breast carcinogenesis and further discuss the prognostic and possible therapeutic impact of pH dysregulation. Please also see the responses to the individual reviewer comments below.

2) There are several contradictory and confusing conclusions in the manuscript remaining to be discussed. It would be better to reorganize the manuscript in a clearer and more convincing manner to present the data in the study.

To better convey the methodological approach and conclusions of the study and avoid confusion, we have now improved the illustration in Figure 1B and included additional schematic illustrations as a new Figure 11.

Additionally, we have reorganized the supplementary material as figure supplements linked up to the regular figures in order to create a clearer structure.

We have also expanded the discussion very substantially (l. 452-469 and 500-565). Particularly, we have added more background and explanations in the areas highlighted as confusing or contradictory by the reviewers. With the included additional information, we hope the editors and reviewers agree that the findings and conclusions of the manuscript are now more convincingly presented.

For more details, see the responses to the individual reviewer comments below.

Reviewer #1:The authors launched an observational study on 110 breast cancer organoids, measured the cancer and normal cell net acid extrusion capacity to correlate different clinicopathological parameters with intracellular pH regulator (NHE1 and NBCn1) level, and unravel the relationship between them and tumor proliferation, lymph node metastasis together with the impact of cotransporter mRNA on patient's survival, thus highlighting the role of pH regulator in malignancy. This study was performed on a large-scale cohort based on tissue samples to comprehensively reflect the heterogeneity setting. However, there lacks detailed mechanism or therapeutic strategies description regarding pH dysregulation. The data has been sufficient to hold back the views proposed and the results will provide better understanding for the role of dysregulated pH in cancer progression and help future studies regarding targeting pH regulator in managing cancer.1. In Figure 9 and 10, SLC9A1 and SLC4A7 mRNA level was analyzed for their correlation with survival outcome, it would be better to supplement survival analysis regarding the expression of SLC9A1 and SLC4A7 protein based on your study cohort.

We appreciate this suggestion. However, even though our cohort used for protein expression analyses and pH measurements – to our knowledge – is the largest existing with functional data on acid-base regulation for any type of human cancer, at this duration (2-5 years) of follow-up, it is not large enough for survival outcome analyses. The number of fatalities is too low to provide the statistical strength needed, and we have now included a comment regarding this limitation in the discussion (l. 540-546).

2. Please specify the reasons for ER and Ki-67 cut-off value setting and include them in Method or insert supporting reference literature.

This is an important point, which we now address on l. 651-661 and 206-212 of the revised manuscript:

Regarding ER (l. 652-656 and 206-212): “We observed a clear distinction between one group of patients with very high estrogen receptor expression (≥90% positive cells) and another group of patients with low estrogen receptor expression (≤15% positive cells) in the breast tumors. […] The absence of intermediate expression levels provided a clear and obvious cut-off value for separation between the two groups.”

Regarding Ki67 (l. 656-661): “As previously noted by others (reference 74), Ki67 expression displays a continuous distribution with no clear separation between groups of high and low expression. The median Ki67 index in our study cohort was 20%; and we therefore followed the guidelines of the 2015 St Gallen International Expert Consensus (reference 74) that in a similar population recommend a cut-off setting of 30% for identification of patients with a clearly high Ki67 index.”

3. It is described in lines 320-321 that the protein expression level of NHE1 and NBCn1 was not significantly influenced by estrogen receptor status. And it is described in lines 333-335 that the protein expression levels of NHE1 as well as NBCn1 were elevated in breast carcinomas with HER2 overexpression or gene amplification both before and after adjustment for other clinicopathological characteristics. But the survival figures of Figure 9, 10 indicated that 'For patients with HER2-enriched breast cancer, we saw no association of SLC4A7 or SLC9A1 mRNA expression with survival; The high overall SLC4A7 expression level in Luminal A breast cancer suggests a prominent role of NBCn1 in this breast cancer molecular subtype. And patients with high SLC9A1 mRNA expression suffering from Luminal A breast cancer showed significantly improved survival (hazard ratio 0.552) compared to patients with low SLC9A1 mRNA expression'. The conclusions seem to be contradictory, please explain this.

The point raised by the reviewer is important and underscores that there is not always straightforward proportionality between mRNA, protein, and function. This is particularly important, when making comparisons across molecular subtypes or clinicopathological characteristics driven by distinct carcinogenic mechanisms. The relationship between mRNA, protein, and function is likely much simpler when compared between individuals within the same molecular subtype; and therefore, we focus primarily on survival analyses performed following molecular subtype stratification (Figure 9, 10 and Figure 10—figure supplement 1 and 2). We now discuss this point on l. 317-324 of the revised manuscript.

The impact of estrogen and HER2 receptors on protein expression is described in Figure 4-6, and we have now included additional analyses to evaluate how *ESR1*, *PGR*, and *ERBB2* expression affects *SLC4A7* and *SLC9A1* mRNA levels (Figure 9-Source Data 1 and Figure 10-Source Data 1). The lower *SLC4A7* mRNA but higher NBCn1 protein level in HER2-enriched breast cancer imply that the HER2-dependent regulation occurs at post-transcriptional level (i.e., by an increased translational activity or higher protein stability). This relationship between mRNA and protein expression for NBCn1 is in line with previous findings in murine breast cancer tissue induced by HER2 overexpression (reference 16), where the *SLC4A7* mRNA level was found low despite an elevated NBCn1 protein level. The predominant post-transcriptional regulation of NBCn1 expression by HER2 complicates the interpretation of the corresponding survival analysis (Figure 10F), and we have now included a comment regarding this point in the manuscript (l. 496-506).

The high *SLC4A7* mRNA level in Luminal A breast cancer is consistent with a prominent regulation of NBCn1 expression at transcriptional level in this molecular subtype; and in congruence, the *SLC4A7* mRNA level carries predictive prognostic value for patients with Luminal A breast cancer (Figure 10D). We have now clarified and expanded this point on l. 346-351 of the revised manuscript.

4. In lines 75-81：'In human and murine breast cancer tissue analyzed without stratification by molecular subtype, Na^+^, HCO_3_^-^ cotransport activity is elevated and protein expression of the NBCn1 (SLC4A7) and MCT1 (SLC16A1) and MCT4 (SLC16A3) are upregulated compared to normal breast tissue. Protein expression of the NHE1 (SLC9A1) is either unchanged or moderately upregulated when breast cancer tissue as one unstratified group is compared to normal breast tissue.' However, worse prognosis was observed in patients suffering from Luminal A or Basal-like/triple-negative breast cancer with high SLC4A7 and/or low SLC9A1 mRNA expression. The expression level of NHE1 (SLC9A1) and the prognostic data are conflicting, please discuss it in detail.

We have now included a much more detailed discussion of NHE1 expression in breast cancer tissue on l. 452-469 of the revised manuscript. Our data establish NHE1 as a metastasis suppressor gene in human breast cancer. Although the spatio-temporal relationships for NHE1 expression need further examination in future studies, it is not unusual that metastasis-suppressor genes are upregulated during early cancer development but downregulated during cancer progression thereby promoting cancer cell metastasis to distant regions (references 49 and 50). We have now expanded the discussion of how NHE1 expression is influenced during breast carcinogenesis and included the existing knowledge regarding NHE1 expression changes during breast cancer progression (l. 452-465). With this additional information, we hope the reviewer agrees that the conclusions based on expression, function, and survival are clearer and more consistent.

We also now emphasize (l. 465-469) that our findings are in accordance with previous studies based on the human MCF7 cancer cell line where NHE1 is upregulated in response to HER2 overexpression (reference 51) but inhibits cell migration (reference 38).

5. The authors found that the expressions of NBCn1 and NHE1 were related to lymph node metastasis and cancer cell proliferation. However, lymph node metastasis and cancer cell proliferation are both key factors predicting worse prognosis. The existing data cannot indicate whether intratumor pH directly affects the patient's prognosis, or indirectly affects the prognosis through lymph nodes metastasis or cancer cell proliferation. Please add survival charts of patients in different subgroups to illustrate it.

We have now included a much more detailed discussion of NHE1 expression in breast cancer tissue on l. 452-469 of the revised manuscript. Our data establish NHE1 as a metastasis suppressor gene in human breast cancer. Although the spatio-temporal relationships for NHE1 expression need further examination in future studies, it is not unusual that metastasis-suppressor genes are upregulated during early cancer development but downregulated during cancer progression thereby promoting cancer cell metastasis to distant regions (references 49 and 50). We have now expanded the discussion of how NHE1 expression is influenced during breast carcinogenesis and included the existing knowledge regarding NHE1 expression changes during breast cancer progression (l. 452-465). With this additional information, we hope the reviewer agrees that the conclusions based on expression, function, and survival are clearer and more consistent.

We also now emphasize (l. 465-469) that our findings are in accordance with previous studies based on the human MCF7 cancer cell line where NHE1 is upregulated in response to HER2 overexpression (reference 51) but inhibits cell migration (reference 38).

6. The authors separately verified the effects of NBCn1 (SLC4A7), MCT1 (SLC16A1), MCT4 (SLC16A3) and NHE1 (SLC9A1) on the prognosis of patients, but whether there exist synergistic or antagonistic effect among the four genes, or whether there is mutual regulation relationship should be addressed.

Our meta-analysis covers data from seven different original studies. Because of variation in the genes covered by the different studies, the n-value drops drastically if only studies covering all four genes are included in a combined analysis of synergism/antagonism. Instead, as suggested by the reviewer, we now examine whether there is mutual regulation relationships between the four genes. This can be done in pair-wise correlation analyses that maintain the original high n-values. As shown in Figure 9-Source Data 1, Figure 10-Source Data 1, Figure 10—figure supplement 1-Source Data 1, and Figure 10—figure supplement 2-Source Data 1, the gene expression levels of the four acid-base transporters vary in response to differences in *ERBB2*, *ESR1*, and *PGR* expression – but when corrected for these effects, we find no independent correlation between the expression levels for any pair of transporters. These findings – described and discussed in the revised manuscript (l. 380-387) – support that the survival effects of the individual transporters are independent.

7. This manuscript requires a more comprehensive summary regarding previous reported role of abnormal pH in promoting cancer in Introduction, and the same about mechanism description and future interfering strategies in Discussion to help readers comprehend this field more conveniently.

Our meta-analysis covers data from seven different original studies. Because of variation in the genes covered by the different studies, the n-value drops drastically if only studies covering all four genes are included in a combined analysis of synergism/antagonism. Instead, as suggested by the reviewer, we now examine whether there is mutual regulation relationships between the four genes. This can be done in pair-wise correlation analyses that maintain the original high n-values. As shown in Figure 9-Source Data 1, Figure 10-Source Data 1, Figure 10—figure supplement 1-Source Data 1, and Figure 10—figure supplement 2-Source Data 1, the gene expression levels of the four acid-base transporters vary in response to differences in *ERBB2*, *ESR1*, and *PGR* expression – but when corrected for these effects, we find no independent correlation between the expression levels for any pair of transporters. These findings – described and discussed in the revised manuscript (l. 380-387) – support that the survival effects of the individual transporters are independent.

8. Breast cancer organoids are mainly composed of cytokeratin 19-positive epithelial cells while the authors only provide the microscopic pictures. It would be better to provide immunofluorescence or IHC staining pictures of cytokeratin 19-positive epithelial cells.

Our meta-analysis covers data from seven different original studies. Because of variation in the genes covered by the different studies, the n-value drops drastically if only studies covering all four genes are included in a combined analysis of synergism/antagonism. Instead, as suggested by the reviewer, we now examine whether there is mutual regulation relationships between the four genes. This can be done in pair-wise correlation analyses that maintain the original high n-values. As shown in Figure 9-Source Data 1, Figure 10-Source Data 1, Figure 10—figure supplement 1-Source Data 1, and Figure 10—figure supplement 2-Source Data 1, the gene expression levels of the four acid-base transporters vary in response to differences in *ERBB2*, *ESR1*, and *PGR* expression – but when corrected for these effects, we find no independent correlation between the expression levels for any pair of transporters. These findings – described and discussed in the revised manuscript (l. 380-387) – support that the survival effects of the individual transporters are independent.

9. Several references date 20 years ago, new references supporting the authors' findings would strengthen the hypothesis. Meanwhile, please arrange it in accordance with the reference format required by eLife.

We have now included additional, newer references. The references dating back more than 20 years now comprise less than 12% of the total number of references and are restricted to seminal papers that lay the theoretical or technical groundwork for the research field.

Our impression – based on the instructions to authors – was that no special reference format is required by *eLife*. We apologize if we have misunderstood this point and now use a reference format that specifies all authors on each paper. We would appreciate further instructions if a different reference format is more convenient – in which case, we will happily modify.

Reviewer #2:In this extensive study, authors have shown that Na^+^,HCO_3_^-^ cotransport and Na^+^/H^+^-exchange transporters predominantly mediate overall cellular acid extrusion in breast cancer microenvironment. In addition, authors demonstrate that higher activity of Na+,HCO_3_^-^-cotransport correlates with increased growth and lymph node metastasis of breast cancer. The mechanistic insights gained from the study will help in the development of therapeutic strategies for breast cancers. The strengths of this study include the use of organoid culture system, and detailed cellular acid-base homeostasis analyses in an extensive cohort of breast cancer patients. Overall, this study is of interest to the peers working in the breast cancer field.1. It would be interesting if the authors have used genetic knockdown or specific inhibitors against NBCn1 and NHE1 in breast cancer organoids to avoid the redundant functions of other ion exchangers.

It is important to note that the current study is based on freshly isolated organoids that are studied after only the delay (hours) needed for enzymatic digestion of the human tissue biopsies. Since the half-lives of protein degradation for NBCn1 and NHE1 in MCF7 human breast cancer cells are around 76 and 48 hours (reference 67), respectively, genetic knockdown techniques would not be possible within the time available. This is now described on l. 551-558 of the revised manuscript. We apologize if the freshly isolated organoids used in this manuscript and their difference from organoid cultures used in other studies were inadequately described. The organoids in our studies were investigated directly after enzymatic isolation without tissue culture because it is well known that culture processes dramatically influence acid-base transporter expression and function. Had the tissue been cultured, it is very likely that it would no longer resemble the human clinical condition. To further emphasize this important point, we have now expanded Figure 1B to include an ultra-brief illustration of the methodological approach and expanded the description on l. 602-603, 125-127, and 551-554 of the revised manuscript.

Regarding specific inhibitors, no such compounds currently exist for NBCn1, which we now mention on l. 558-559 of the revised manuscript. We have previously shown that the Na^+^,HCO_3_^–^-cotransport in human breast cancer tissue has low sensitivity to 4,4′-diisothiocyano-2,2′-stilbenedisulfonic acid (DIDS), which is a pharmacological characteristic of NBCn1 compared to other Na^+^,HCO_3_^–^-cotransporters. A comment and reference to this information has been included on l. 559-565 of the revised manuscript.

2. Did the authors analyze the expression of angiogenic markers such as CD31 or CD34 in the patient samples?

For our cohort evaluated with pH dynamics and protein expression levels for NBCn1 and NHE1, we did not analyze the expression of angiogenic markers. In addition to the protein expression analyses for NBCn1 and NHE1, the pathological information stems from the standard diagnostic pathology evaluation, which did not include CD31/PECAM or CD34. However, to accommodate the request from the reviewer, we have now assessed how CD31 and CD34 mRNA expression relates to the different breast cancer molecular subtypes evaluated in the meta-analysis. These data are now included in Figure 9—figure supplement 1 and mentioned on l. 331-333 of the revised manuscript.

3. For the ease of readers, in Figure 1B, authors should also highlight the observations on expression and correlation of NBCn1 and NHE1 with overall survival in different breast cancer subtypes.

This is a point well taken. Instead of expanding Figure 1B – which has already been expanded with methodological details, see point 1 above – we have included new summary schematics of our conclusions regarding NHE1 and NBCn1 in a new Figure 11 in order to best maintain the natural progression of the manuscript.

4. Authors should discuss the possible limitations of their study in the Discussion part of the manuscript.

We now discuss limitations of the study on l. 540-550 of the revised manuscript. These include considerations regarding n-values and follow-up time in our cohort. We also discussion that the sizes of individual subgroups are determined by the natural incidence of the various clinicopathological characteristics, which results in a less balanced distribution for some breast cancer characteristics.

On l. 551-565, we also emphasize that the current approach based on freshly isolated organoids provides the closest possible evaluation of in vivo characteristics under conditions where experimental buffer compositions can be modified and controlled. In order to study detailed molecular mechanisms and their influence on the cellular biology, future studies in model systems will be important.

5. Please correct the spelling of Maier to Kaplan-Meier in methods.

Thank you for pointing this out – corrected twice in the methods section (l. 678 and 700).

6. In Figure 1 E, authors should use only one asterisk to show the statistical significance between the groups.

The asterisks were meant to indicate the statistical difference relating to two different comparisons between the cancer tissue and the normal breast tissue: one in the presence of CO_2_/HCO_3_^–^, the other in the absence of CO_2_/HCO_3_^–^. We apologize if this was not clear and have now changed the layout of the figure panel to avoid confusion.

Reviewer #3:Nicolai J. Toft et al. investigated the feature of acid-base metabolism in breast cancer tissue from several aspects, revealing the underlying mechanism of pH regulation according to the heterogeneity. Moreover, the acid-base transport and intracellular pH state were proved capable of predicting the proliferation ability and lymph node metastasis risk of breast cancer, and were associated with the survival in specific breast cancer subtypes. The conclusions of this paper are mostly well supported by data. However, it would have been clearer and more convincing to present the clinical significance combining all data in this study, especially in a more direct way (using tables or graphs).1. An exposition for combined analysis of the results in figure 1-6 and figure 7-8 is expected to let the readers better understand the clinical significance (like prediction for proliferation and lymph node metastasis) of the various situations in breast cancer with different acid-base metabolism features.

As suggested, we have now markedly expanded the explanation and discussion of the analyses on l. 452-469, and 507-565 of the revised manuscript. The additional discussion highlights the clinical significance and several pathological characteristics in breast cancer that based on our findings are likely to be at least partly due to the acid-base features of the individual molecular subtypes.

2. There are some questions of how to select different indicators to evaluate the prognosis tendency. For example, data in figure 5E-F and 6E-F showed that Her2 receptor increased the expression of both NBCn1 and NHE1, which exert opposite predictive function in lymph node metastasis according to the data in figure 8G. In this condition, how to evaluate the risk of metastasis? What is the clinical significance of the data on Her2-enriched subtype?

This is an excellent point, which we now discuss in greater detail on l. 507-514 of the revised manuscript. The dual effect of HER2 on expression of NBCn1 and NHE1 is intriguing as it suggests that more detailed analyses (e.g., relative regulation of NBCn1 vs NHE1) may provide more accurate predictive value than simple evaluation of HER2 overexpression/gene amplification. Our findings furthermore suggest that selective targeting of individual HER2-activated downstream signaling pathways (including individual acid-base transporters) could optimize the current therapeutic approach based on direct HER2 inhibition.

3. How to confirm the statistical significance of the difference in the figures showing "traces of NH_4_^+^-prepulse-induced pH_i_ dynamics" (like figure 2A-B)? Meanwhile, the p value is better to be marked in the figures.

We have now included a paragraph (l. 151-155) that explains the relationship between the different panels, i.e., how data extracted from the pH_i_ traces (like Figure 2A,B) were summarized and statistically compared in the associated panels (like Figure 2C,D and Figure 2—figure supplement 2). The statistical analyses compared the net acid extrusion capacities (Figure 2C) and steady-state pH_i_ levels (Figure 2D) derived from the pH_i_ traces. We include the traces in order to give the reader a better feel for the recordings underlying our analyses, which is in line with the recommendation in the *eLife* instructions to authors that traces should be included whenever possible.

4. The standard to define estrogen receptor-negative breast cancer and estrogen receptor-positive breast cancer should be pointed out. The results in figure 4 is confusing.

We apologize if the description of the nomenclature was inadequate and confusing. We have now described in detail how we distinguished between breast cancer samples with highly elevated estrogen receptor expression (ER-positive breast cancer: ≥90% ER^+^ cells) and samples where the estrogen receptor expression was in a range similar to that previously reported for normal breast tissue (ER-normal breast cancer: ≤15% ER^+^ cells) on l. 206-212 and 652-656 of the revised manuscript.

5. The information in figure 10A and 10F showed that the mRNA of SLC4A7, which encodes protein NBCn1, expressed in a very low level and could not predict the survival prognosis in HER2-riched breast cancer subtype. However, in figure 5F and figure 6F, the data indicated that HER2 expression level was positively associated with higher NBCn1 frequency. So, a further explanation is needed for the contradiction of these data and their clinical significance.

This is a very good point. The lower *SLC4A7* mRNA but higher NBCn1 protein level in HER2-enriched breast cancer imply that the HER2-dependent regulation occurs at post-transcriptional level (i.e., by an increased translational activity or higher protein stability). This relationship between mRNA and protein expression for NBCn1 is in line with previous findings in murine breast cancer tissue induced by HER2 overexpression (reference 16), where the *SLC4A7* mRNA level was found low despite an elevated NBCn1 protein level. The predominant post-transcriptional regulation of NBCn1 expression by HER2 complicates the interpretation of the corresponding survival analysis (Figure 10F), and we have now included a comment regarding this point in the manuscript (l. 496-506).

6. The authors should add more biological and molecular experiments to better support the conclusions.

The unique strength of the current study is the clinically relevant functional data on pH dynamics from a human cohort of previously unparalleled size. The data set is unique because it so clearly links up to the human clinical condition and embraces its heterogeneity and complexity, thereby allowing us to establish relationships between pH dynamics and clinical and pathological characteristics, which has not been possible before. To establish this link convincingly, we performed the functional recordings immediately after organoid isolation from the sampled tissue biopsies in order to avoid culture-induced phenotypical changes. The short time period between tissue sampling and the functional experiments, however, makes the approach unsuited for molecular interventions as the half-lives of protein degradation is ~76 hours for NBCn1 and ~48 hours for NHE1 (reference 81). We now discuss these points in the revised manuscript (l. 551-565).

Although beyond the scope of the current study, the results from the current study has already prompted new biological and molecular investigations in our group based on murine breast cancer models that are further away from the human clinical condition but much better suited for molecular intervention. Undoubtedly, the current study will also stimulate new molecular studies from colleagues in the field.